# CONVERGENT AND EFFICIENT DEEP Q NETWORK ALGORITHM

**Zhikang T. Wang & Masahito Ueda**[*]
Department of Physics and Institute for Physics of Intelligence
University of Tokyo
7-3-1 Hongo, Bunkyo-ku, Tokyo 113-0033, Japan
`{wang,ueda}@cat.phys.s.u-tokyo.ac.jp`

## ABSTRACT

Despite the empirical success of the deep Q network (DQN) reinforcement learning algorithm and its variants, DQN is still not well understood and it does not guarantee convergence. In this work, we show that DQN can indeed diverge and cease to operate in realistic settings. Although there exist gradient-based convergent methods, we show that they actually have inherent problems in learning dynamics which cause them to fail even in simple tasks. To overcome these problems, we propose a convergent DQN algorithm (*C-DQN*) that is guaranteed to converge and can work with large discount factors ($\sim 0.9998$). It learns robustly in difficult settings and can learn several difficult games in the Atari 2600 benchmark that DQN fails to solve.

## 1 INTRODUCTION

With the development of deep learning, reinforcement learning (RL) that utilizes deep neural networks has demonstrated great success recently, finding applications in various fields including robotics, games, and scientific research (Levine et al., 2018; Berner et al., 2019; Fösel et al., 2018; Wang et al., 2020). One of the most efficient RL strategy is Q-learning (Watkins, 1989), and the combination of Q-learning and deep learning leads to the DQN algorithms (Mnih et al., 2015; Hessel et al., 2018; Riedmiller, 2005), which hold records on many difficult RL tasks (Badia et al., 2020). However, unlike supervised learning, Q-learning, or more generally temporal difference (TD) learning, does not guarantee convergence when function approximations such as neural networks are used, and as a result, their success is actually empirical, and the performance relies heavily on hyperparameter tuning and technical details involved. This happens because the agent uses its own prediction to construct the learning objective, a.k.a. bootstrapping, and as it generalizes, its predictions over different data interfere with each other, which can make its learning objective unstable in the course of training and potentially lead to instability and divergence.

This non-convergence problem was pointed out decades ago by the pioneering works of Baird (1995) and Tsitsiklis & Van Roy (1997), and it has been empirically investigated for DQN by Van Hasselt et al. (2018). We have also observed the divergence of DQN in our experiments, as in Fig. 6. The non-convergence problem often shows up as instability in practice and it places significant obstacles to the application of DQN to complicated tasks. It makes the training with deeper neural networks more difficult, limits the time horizon for planning, and makes the results sometimes unstable and sensitive to hyperparameters. This state of affairs is not satisfactory especially for those scientific applications that require convergence and generality. Although convergent gradient-based methods have also been proposed (Sutton et al., 2009; Bhatnagar et al., 2009; Feng et al., 2019; Ghiassian et al., 2020), they cannot easily be used with deep non-linear neural networks as they either require linearity or involve computationally heavy operations, and they often show worse empirical performance compared with TD methods.

In this work, we show that the above-mentioned gradient-based methods actually have inherent problems in learning dynamics which hamper efficient learning, and we propose a convergent DQN

---

[*]RIKEN Center for Emergent Matter Science (CEMS), Wako, Saitama 351-0198, Japan

(*C-DQN*) algorithm by modifying the loss of DQN. Because an increase of loss upon updating the target network of DQN is a necessary condition for its divergence, we construct a loss that does not increase upon the update of the target network, and therefore, the proposed algorithm converges in the sense that the loss monotonically decreases. In Sec. 2 we present the background. In Sec. 3 we discuss the inefficiency issues in the previous gradient-based methods and demonstrate using toy problems. In Sec. 4 we propose C-DQN and show its convergence. In Sec. 5 we show the results of C-DQN on the *Atari 2600* benchmark (Bellemare et al., 2013) and in Sec. 6 we present the conclusion and future prospect. To our knowledge, the proposed C-DQN algorithm is the first convergent RL method that is sufficiently efficient and scalable to obtain successful results on the standard *Atari 2600* benchmark using deep neural networks, showing its efficacy in dealing with realistic and complicated problems.

## 2 BACKGROUND

Reinforcement learning involves a Markov decision process (MDP), where the state $s_t$ of an environment at time step $t$ makes a transition to the next state $s_{t+1}$ conditioned on the action of the agent $a_t$ at time $t$, producing a reward $r_t$ depending on the states. The process can terminate at terminal states $s_T$, and the transition of states can be either probabilistic or deterministic. The goal is to find a policy $\pi(s)$ to determine the actions $a_{t+i} \sim \pi(s_{t+i})$ in order to maximizes the return $\sum_{i=0}^{T-t} r_{t+i}$, i.e., the sum of future rewards. In practice, a discounted return $\sum_{i=0}^{T-t} \gamma^i r_{t+i}$ is often used instead, with the discount factor $\gamma < 1$ and $\gamma \approx 1$, so that the expression is convergent for $T \to \infty$ and that rewards far into the future can be ignored, giving an effective time horizon $\frac{1}{1-\gamma}$. The value function is defined as the expected return for a state $s_t$ following a policy $\pi$, and the Q function is defined as the expected return for a state-action pair $(s_t, a_t)$:

$$V_\pi(s_t) = \mathbb{E}_{a_t, \{(s_{t+i}, a_{t+i})\}_{i=1}^{T-t}} \left[ \sum_{i=0}^{T-t} \gamma^i r_{t+i} \right], \quad Q_\pi(s_t, a_t) = \mathbb{E}_{\{(s_{t+i}, a_{t+i})\}_{i=1}^{T-t}} \left[ \sum_{i=0}^{T-t} \gamma^i r_{t+i} \right], \quad (1)$$

with $a_{t+i} \sim \pi(s_{t+i})$ in the evaluation of the expectation. When the Q function is maximized by a policy, we say that the policy is optimal and denote the Q function and the policy by $Q^*$ and $\pi^*$, respectively. The optimality implies that $Q^*$ satisfies the Bellman equation (Sutton & Barto, 2018)

$$Q^*(s_t, a_t) = r_t + \gamma \mathbb{E}_{s_{t+1}} \left[ \max_{a'} Q^*(s_{t+1}, a') \right]. \quad (2)$$

The policy $\pi^*$ is greedy with respect to $Q^*$, i.e. $\pi^*(s) = \arg\max_{a'} Q^*(s, a')$. Q-learning uses this recursive relation to learn $Q^*$. In this work we only consider the deterministic case and drop the notation $\mathbb{E}_{s_{t+1}}[\cdot]$ where appropriate.

When the space of state-action pairs is small and finite, we can write down the values of an arbitrarily initialized Q function for all state-action pairs into a table, and iterate over the values using

$$\Delta Q(s_t, a_t) = \alpha \left( r_t + \gamma \max_{a'} Q(s_{t+1}, a') - Q(s_t, a_t) \right), \quad (3)$$

where $\alpha$ is the learning rate. This is called Q-table learning and it guarantees convergence to $Q^*$. If the space of $(s, a)$ is large and Q-table learning is impossible, a function approximation is used instead, representing the Q function as $Q_\theta$ with learnable parameter $\theta$. The learning rule is

$$\Delta \theta = \alpha \nabla_\theta Q_\theta(s_t, a_t) \left( r_t + \gamma \max_{a'} Q_\theta(s_{t+1}, a') - Q_\theta(s_t, a_t) \right), \quad (4)$$

which can be interpreted as modifying the value of $Q_\theta(s_t, a_t)$ following the gradient so that $Q_\theta(s_t, a_t)$ approaches the target value $r_t + \gamma \max_{a'} Q_\theta(s_{t+1}, a')$. However, this iteration may not converge, because the term $\max_{a'} Q_\theta(s_{t+1}, a')$ is also $\theta$-dependent and may change together with $Q_\theta(s_t, a_t)$. Specifically, an exponential divergence occurs if $\gamma \nabla_\theta Q_\theta(s_t, a_t) \cdot \nabla_\theta \max_{a'} Q_\theta(s_{t+1}, a') > ||\nabla_\theta Q_\theta(s_t, a_t)||^2$ is always satisfied and the value of $\max_{a'} Q_\theta(s_{t+1}, a')$ is not constrained by other means.[1] This can be a serious issue for realistic tasks, because the adjacent states $s_t$ and $s_{t+1}$ often have similar representations and $\nabla_\theta Q_\theta(s_t, \cdot)$ is close to $\nabla_\theta Q_\theta(s_{t+1}, \cdot)$.

---

[1]This can be shown by checking the Bellman error $\delta_t := r_t + \gamma \max_{a'} Q_\theta(s_{t+1}, a') - Q_\theta(s_t, a_t)$, for which we have $\Delta \delta_t = \alpha \delta_t \left( \gamma \nabla_\theta Q_\theta(s_t, a_t) \cdot \max_{a'} Q_\theta(s_{t+1}, a') - ||\nabla_\theta Q_\theta(s_t, a_t)||^2 \right)$ up to the first order of $\Delta \theta$ following Eq. (4). As $\Delta \delta_t$ is proportional to $\delta_t$ with the same sign, it can increase exponentially.

The DQN algorithm uses a deep neural network with parameters $\theta$ as $Q_\theta$ (Mnih et al., 2015), and to stabilize learning, it introduces a target network with parameters $\tilde{\theta}$, and replace the term $\max_{a'} Q_\theta(s_{t+1}, a')$ by $\max_{a'} Q_{\tilde{\theta}}(s_{t+1}, a')$, so that the target value $r_t + \gamma \max_{a'} Q_{\tilde{\theta}}(s_{t+1}, a')$ does not change simultaneously with $Q_\theta$. The target network $\tilde{\theta}$ is then updated by copying from $\theta$ for every few thousand iterations of $\theta$. This technique reduces fluctuations in the target value and dramatically improves the stability of learning, and with the use of offline sampling and adaptive optimizers, it can learn various tasks such as video games and simulated robotic control (Mnih et al., 2015; Lillicrap et al., 2015). Nevertheless, the introduction of the target network $\tilde{\theta}$ is not well-principled, and it does not really preclude the possibility of divergence. As a result, DQN sometimes requires a significant amount of hyperparameter tuning in order to work well for a new task, and in some cases, the instability in learning can be hard to diagnose or remove, and usually one cannot use a discount factor $\gamma$ that is very close to 1. In an attempt to solve this problem, Durugkar & Stone (2017) considered only updating $\theta$ in a direction that is perpendicular to $\nabla_\theta \max_{a'} Q_\theta(s_{t+1}, a')$; however, this strategy is not satisfactory in general and can lead to poor performance, as shown in Pohlen et al. (2018).

One way of approaching this problem is to consider the mean squared Bellman error (MSBE), which is originally proposed by Baird (1995) and called the *residual gradient* (RG) algorithm. The Bellman error, or Bellman residual, TD error, is given by $\delta_t(\theta) := r_t + \gamma \max_{a'} Q_\theta(s_{t+1}, a') - Q_\theta(s_t, a_t)$. Given a dataset $\mathcal{S}$ of state-action data, $\delta_t$ is a function of $\theta$, and we can minimize the MSBE loss

$$L_{MSBE}(\theta) := \mathbb{E}\left[|\delta(\theta)|^2\right] = \frac{1}{|\mathcal{S}|} \sum_{(s_t, a_t, r_t, s_{t+1}) \in \mathcal{S}} \left|Q_\theta(s_t, a_t) - r_t - \gamma \max_{a'} Q_\theta(s_{t+1}, a')\right|^2, \quad (5)$$

and in practice the loss is minimized via gradient descent. If $L_{MSBE}$ becomes zero, we have $\delta_t \equiv 0$ and the Bellman equation is satisfied, implying $Q_\theta = Q^*$. Given a fixed dataset $\mathcal{S}$, the convergence of the learning process simply follows the convergence of the optimization of the loss. This strategy can be used with neural networks straightforwardly. There have also been many improvements on this strategy including Sutton et al. (2008; 2009); Bhatnagar et al. (2009); Dai et al. (2018); Feng et al. (2019); Ghiassian et al. (2020); Touati et al. (2018), and they are often referred to as gradient-TD methods. Many of them have focused on how to evaluate the expectation term in Eq. (2) and make it converge to the same solution found by TD, or Q-learning. However, most of these methods often do not work well for difficult problems, and few of them have been successfully demonstrated on standard RL benchmarks, especially the *Atari 2600* benchmark. In the following we refer to the strategy of simply minimizing $L_{MSBE}$ as the RG algorithm, or RG learning.

## 3   INEFFICIENCY OF MINIMIZING THE MSBE

### 3.1   ILL-CONDITIONNESS OF THE LOSS

In the following we show that minimizing $L_{MSBE}$ may not lead to efficient learning by considering the case of deterministic tabular problems, for which all the gradient-TD methods mentioned above reduce to RG learning. For a tabular problem, the Q function values for different state-action pairs can be regarded as independent variables. Suppose we have a trajectory of experience $\{(s_t, a_t, r_t)\}_{t=0}^{N-1}$ obtained by following the greedy policy with respect to $Q$, i.e. $a_t = \arg\max_{a'} Q(s_t, a')$, and $s_N$ is a terminal state. Taking $\gamma = 1$, the MSBE loss is given by

$$L_{MSBE} = \frac{1}{N}\left[\sum_{t=0}^{N-2}(Q(s_t, a_t) - r_t - Q(s_{t+1}, a_{t+1}))^2 + (Q(s_{N-1}, a_{N-1}) - r_{N-1})^2\right]. \quad (6)$$

Despite the simple quadratic form, the Hessian matrix of $L_{MSBE}$ with variables $\{Q(s_t, a_t)\}_{t=0}^{N-1}$ is ill-conditioned, and therefore does not allow efficient gradient descent optimization. The condition number $\kappa$ of a Hessian matrix is defined by $\kappa := \frac{|\lambda_{max}|}{|\lambda_{min}|}$, where $\lambda_{max}$ and $\lambda_{min}$ are the largest and smallest eigenvalues. We have numerically found that $\kappa$ of the Hessian of $L_{MSBE}$ in Eq. (6) grows as $O(N^2)$. To find an analytic expression, we add an additional term $Q(s_0, a_0)^2$ to $L_{MSBE}$, so that the

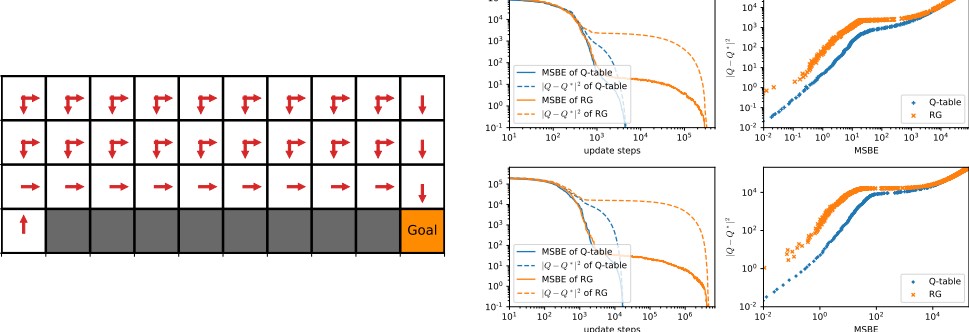

Figure 1: Left: the cliff walking task, where the agent is supposed to go to the lower right corner as quickly as possible and avoid the grey region. The red arrows show the optimal policy. In this example the system has the height of 4 and the width of 10. Right: results of learning the cliff walking task in a tabular setting, using randomly sampled state-action pair data. The upper plots show the result with width 10 and $\gamma = 0.9$, and the lower show the result with width 20 and $\gamma = 0.95$. $|Q - Q^*|^2$ is the squared distance between the learned Q function and the optimal $Q^*$.[3] Both Q-table learning and RG use the learning rate of $0.5$, averaged over 10 repeated runs.

Hessian matrix becomes

$$
\begin{pmatrix}
4 & -2 & & \\
-2 & 4 & \ddots & \\
& \ddots & \ddots & -2 \\
& & -2 & 4
\end{pmatrix}_{N \times N} .
\tag{7}
$$

The eigenvectors of this Hessian matrix that have the form of standing waves are given by $(\sin \frac{k\pi}{N+1}, \sin \frac{2k\pi}{N+1}, \dots \sin \frac{Nk\pi}{N+1})^T$ for $k \in \{1, 2, \dots N\}$, and the corresponding eigenvalues are given by $4 - 4\cos\frac{k\pi}{N+1}$. Therefore, we have $\kappa = \frac{1 + \cos\frac{\pi}{N+1}}{1 - \cos\frac{\pi}{N+1}} \sim O(N^2)$. See appendix for the details.

With $\gamma < 1$, if the states form a cycle, i.e., if $s_{N-1}$ makes a transition to $s_0$, the loss becomes $L_{MSBE} = \frac{1}{N}[\sum_{t=0}^{N-2} (Q_t - r_t - \gamma Q_{t+1})^2 + (Q_{N-1} - r_{N-1} - \gamma Q_0)^2]$, where $Q(s_t, a_t)$ is denoted by $Q_t$. Then, the Hessian matrix is cyclic and the eigenvectors have the form of periodic waves: $(\sin \frac{2k\pi}{N}, \sin \frac{4k\pi}{N}, \dots \sin \frac{2Nk\pi}{N})^T$ and $(\cos \frac{2k\pi}{N}, \cos \frac{4k\pi}{N}, \dots \cos \frac{2Nk\pi}{N})^T$ for $k \in \{1, 2, \dots \frac{N}{2}\}$, with corresponding eigenvalues given by $2(1 + \gamma^2) - 4\gamma \cos\frac{2k\pi}{N}$. At the limit of $N \to \infty$, we have $\kappa = \frac{(1+\gamma)^2}{(1-\gamma)^2}$. Using $\gamma \approx 1$, we have $\kappa \sim O\left(\frac{1}{(1-\gamma)^2}\right)$. By interpreting $\frac{1}{1-\gamma}$ as the effective time horizon, or the effective size of the problem, we see that $\kappa$ is quadratic in the size of the problem, which is the same as its dependence on $N$ in the case of $\gamma = 1$ above. In practice, $\kappa$ is usually $10^4 \sim 10^5$, and we therefore conclude that the loss is ill-conditioned.

The ill-conditionedness has two important implications. First, as gradient descent converges at a rate of $O(\kappa)$,[2] the required learning time is quadratic in the problem size, i.e. $O(N^2)$, for the RG algorithm. In contrast, Q-learning only requires $O(N)$ steps as it straightforwardly propagates reward information from $s_{i+1}$ to $s_i$ at each iteration step. Therefore, the RG algorithm is significantly less computationally efficient than Q-learning. Secondly, the ill-conditionedness implies that a small $L_{MSBE}$ does not necessarily correspond to a small distance between the learned $Q$ and the optimal $Q^*$, which may explain why $L_{MSBE}$ is not a useful indicator of performance (Geist et al., 2017).

**Cliff walking**  By way of illustration, we consider a tabular task, the *cliff walking* problem in Sutton & Barto (2018), as illustrated in Fig. 1. The agent starts at the lower left corner in the grid and can move into nearby blocks. If it enters a white block, it receives a reward $-1$; if it enters a grey block which is the cliff, it receives a reward $-100$ and the process terminates; if it

---

[2]Although in the deterministic case momentum can be used to accelerate gradient descent to a convergence rate of $O(\sqrt{\kappa})$, it cannot straightforwardly be applied to stochastic gradient descent since it requires a large momentum factor $(\frac{\sqrt{\kappa}-1}{\sqrt{\kappa}+1})^2$ (Polyak, 1964) which results in unacceptably large noise.

[3]$|Q - Q^*|^2$ is defined by $\sum_{(s,a)} |Q(s,a) - Q^*(s,a)|^2$, where the sum is taken over all state-action pairs.

enters the goal, it terminates with a zero reward. To learn this task, we initialize $Q$ for all state-action pairs to be zero, and we randomly sample a state-action pair as $(s_t, a_t)$ and find the next state $s_{t+1}$ to update $Q$ via Eq. (3), which is the Q-table learning, or to minimize the associated loss $(Q_\theta(s_t, a_t) - r_t - \gamma \max_{a'} Q_\theta(s_{t+1}, a'))^2$ following the gradient, which is RG learning. As shown in Fig. 1, RG learns significantly more slowly than Q-table learning, and as shown in the right plots, given a fixed value of $L_{MSBE}$, RG's distance to the optimal solution $Q^*$ is also larger than that of Q-table learning, showing that Q-table learning approaches $Q^*$ more efficiently. To investigate the dependence on the size of the problem, we consider a width of 10 of the system with $\gamma = 0.9$, and its doubled size—a width of 20 with $\gamma = 0.95$. The results are presented in the upper and lower plots in Fig. 1. Notice that the agent learns by random sampling from the state-action space, and doubling the system size reduces the sampling efficiency by a factor of 2. As the learning time of Q-learning is linear and RG is quadratic in the deterministic case, their learning time should respectively become 4 times and 8 times for the double-sized system. We have confirmed that the experimental results approximately coincide with this prediction and therefore support our analysis of the scaling property above.

## 3.2 TENDENCY OF MAINTAINING THE AVERAGE PREDICTION

Besides the issue of the ill-conditionedness, the update rule of RG learning still has a serious problem which can lead to unsatisfactory learning behaviour. To show this, we first denote $Q_t \equiv Q(s_t, a_t)$ and $Q_{t+1} \equiv \max_{a'} Q(s_{t+1}, a')$, and given that they initially satisfy $Q_t = \gamma Q_{t+1}$, with an observed transition from $s_t$ to $s_{t+1}$ with a non-zero reward $r_t$, repeatedly applying the Q-table learning rule in Eq. (3) leads to $\Delta Q_t = r_t$ and $\Delta Q_{t+1} = 0$, and thus $\Delta (Q_t + Q_{t+1}) = r_t$. On the other hand, in the case of RG, minimizing $L_{MSBE}$ using the gradient results in the following learning rule

$$\Delta Q(s_t, a_t) = \alpha \left( r_t + \gamma \max_{a'} Q(s_{t+1}, a') - Q(s_t, a_t) \right), \quad \Delta \max_{a'} Q(s_{t+1}, a') = -\gamma \Delta Q(s_t, a_t), \tag{8}$$

and therefore whenever $Q_t$ is modified, $Q_{t+1}$ changes simultaneously, and we have

$$\Delta (Q_t + Q_{t+1}) = (1 - \gamma) r_t. \tag{9}$$

Due to the condition $\gamma \approx 1$, $\Delta (Q_t + Q_{t+1})$ can be very small, which is different from Q-learning, since the sum of the predicted $Q$, i.e. $\sum_t Q_t$, almost does not change. This occurs because there is an additional degree of freedom when one modifies $Q_t$ and $Q_{t+1}$ to satisfy the Bellman equation: Q-learning tries to keep $Q_{t+1}$ fixed, while RG follows the gradient and keeps $Q_t + \frac{1}{\gamma} Q_{t+1}$ fixed, except for the case where $s_{t+1}$ is terminal and $Q_{t+1}$ is a constant. This has important implications on the learning behaviour of RG as heuristically explained below.

If the average of $Q(s, a)$ is initialized above that of $Q^*(s, a)$ and the transitions among the states can form loops, the learning time of RG additionally scales as $O(\frac{1}{1-\gamma})$ due to Eq. (9), regardless of the finite size of the problem. As shown in Fig. 2, in the cliff walking task with width 10, Q-table learning has roughly the same learning time for different $\gamma$, while RG scales roughly as $O(\frac{1}{1-\gamma})$ and does not learn for $\gamma = 1$. This is because the policy $\arg\max_{a'} Q(s, a')$ of RG prefers non-terminal states and goes into loops, since the transitions to terminal states are associated with $Q$ values below what the agent initially expects. Then, Eq. (9) controls the learning dynamics of the sum of all $Q$ values, i.e. $\sum_{(s,a)} Q(s, a)$, with the learning target being $\sum_{(s,a)} Q^*(s, a)$, and the learning time scales as $O(\frac{1}{1-\gamma})$. For $\gamma = 1$, $\Delta \sum_{(s,a)} Q_{(s,a)}$ is always zero and $\sum_{(s,a)} Q^*(s, a)$ cannot be learned, which results in failure of learning.

A more commonly encountered failure mode appears when $Q$ is initialized below $Q^*$, in which case the agent only learns to obtain a small amount of reward and faces difficulties in consistently improving its performance. A typical example is shown in Fig. 3, where $Q$ is initialized to be zero and the agent learns from its observed state transitions in an online manner following the $\epsilon$-greedy policy.[4] We see that while Q-table learning can solve the problem easily without the help of a non-zero $\epsilon$, RG cannot find the optimal policy with $\epsilon = 0$, and it learns slowly and relies on non-zero $\epsilon$ values. This is because when RG finds rewards, $Q(s, a)$ for some states increase while the other $Q(s, a)$ values

---

[4]$\epsilon$-greedy means that for probability $\epsilon$ a random action is used; otherwise $\arg\max_{a'} Q(s, a')$ is used.

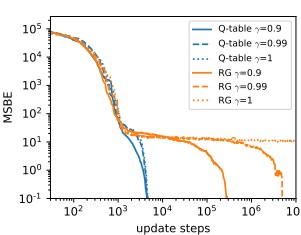
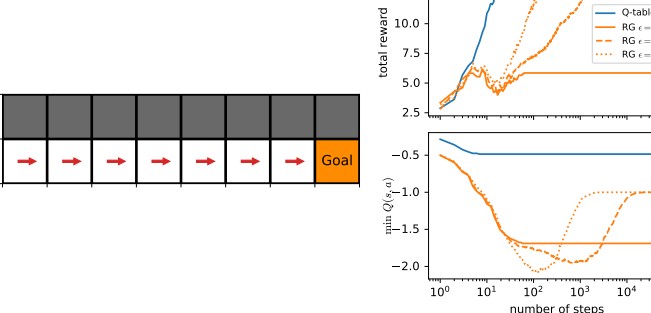

Figure 2: Results of cliff walking in Fig. 1 with different $\gamma$, with system's width of 10, averaged over 10 runs. RG with $\gamma = 1$ does not converge within reasonable computational budgets.

Figure 3: Left: one-way cliff walking task, where the agent starts at the lower left corner, and at each step it is allowed to move to the right to obtain a reward of 2, or move up and terminate with a reward of $-1$. It terminates upon reaching the goal. Right: performance of the learned greedy policy and $\min Q(s, a)$ for online Q-table learning and RG, following the $\epsilon$-greedy policy for different values of $\epsilon$, with $\gamma = 1$ and a learning rate of $0.5$, averaged over 100 trials.

decrease and may become negative, and if the negative values become smaller than the reward associated with the termination, i.e. $-1$, it will choose termination as its best action. Therefore, it relies on the exploration strategy to correct its behaviour at those states with low $Q(s, a)$ values and to find the optimal policy. Generally, when an unexpected positive reward $r_t$ is found in learning, according to the learning rule in Eq. (8), with an increase of $r_t$ in $Q(s_t, a_t)$, $\max_{a'} Q(s_{t+1}, a')$ decreases simultaneously by $r_t$, and the action at $s_{t+1}$, i.e. $\arg\max_{a'} Q(s_{t+1}, a')$, is perturbed, which may make the agent choose a worse action that leads to $r_t$ less reward, and therefore the performance may not improve on the whole. In such cases, the performance of RG crucially relies on the exploration strategy so that the appropriate action at $s_{t+1}$ can be rediscovered. In practice, especially for large-scale problems, efficient exploration is difficult in general and one cannot enhance exploration easily without compromising the performance, and therefore, RG often faces difficulties in learning and performs worse than Q-learning for difficult and realistic problems.

**Remark** Although the above discussion is based on the tabular case, we believe that the situation is not generally better when function approximations are involved. With the above issues in mind, it can be understood why most of the currently successful examples of gradient-TD methods have tunable hyperparameters that can reduce the methods to conventional TD learning, which has a Q-learning-style update rule. If the methods get closer to conventional TD without divergence, they typically achieve better efficiency and better quality of the learned policy. When the agent simply learns from the gradient of the loss without using techniques like target networks, the performance can be much worse. This probably explains why the performance of the PCL algorithm (Nachum et al., 2017) sometimes deteriorates when the value and the policy neural networks are combined into a unified Q network, and it has been reported that the performance of PCL can be improved by using a target network (Gao et al., 2018). Note that although ill-conditionedness may be resolved by a second-order optimizer or the Retrace loss (Munos et al., 2016; Badia et al., 2020), the issue in Sec. 3.2 may not be resolved, because it will likely converge to the same solution as the one found by gradient descent and thus have the same learning behaviour. A rigorous analysis of the issue in Sec. 3.2 is definitely desired and is left for future work.

## 4 CONVERGENT DQN ALGORITHM

### 4.1 INTERPRETING DQN AS FITTED VALUE ITERATION

As we find that Q-learning and the related conventional TD methods and DQN have learning dynamics that is preferable to RG, we wish to minimally modify DQN so that it can maintain its learning dynamics while being convergent. To proceed, we first cast DQN into the form of fitted value iteration (FVI) (Ernst et al., 2005; Munos & Szepesvári, 2008). With initial parameters $\tilde{\theta}_0$ of the target network, the DQN loss for a transition $(s_t, a_t, r_t, s_{t+1})$ and network parameters $\theta$ is defined as

$$\ell_{DQN}(\theta; \tilde{\theta}_i) := \left( Q_\theta(s_t, a_t) - r_t - \gamma \max_{a'} Q_{\tilde{\theta}_i}(s_{t+1}, a') \right)^2, \tag{10}$$

and DQN learns by iterating over the target network

$$L_{DQN}(\theta; \tilde{\theta}_i) := \mathbb{E}\left[l_{DQN}(\theta; \tilde{\theta}_i)\right], \qquad \tilde{\theta}_{i+1} = \arg\min_\theta L_{DQN}(\theta; \tilde{\theta}_i), \qquad (11)$$

and $\tilde{\theta}_i$ is used as the parameter of the trained network for a sufficiently large $i$. In practice, the minimum in Eq. (11) is found approximately by stochastic gradient descent, but for simplicity, here we consider the case where the minimum is exact. When DQN diverges, the loss is supposed to diverge with iterations, which means we have $\min_\theta L_{DQN}(\theta; \tilde{\theta}_{i+1}) > \min_\theta L_{DQN}(\theta; \tilde{\theta}_i)$ for some $i$.

### 4.2 Constructing a non-increasing series

**Theorem 1.** *The minimum of $L_{DQN}(\theta; \tilde{\theta}_i)$ with target network $\tilde{\theta}_i$ is upper bounded by $L_{MSBE}(\tilde{\theta}_i)$.*

This relation can be derived immediately from $L_{DQN}(\tilde{\theta}_i; \tilde{\theta}_i) = L_{MSBE}(\tilde{\theta}_i)$ and $\min_\theta L_{DQN}(\theta; \tilde{\theta}_i) \le L_{DQN}(\tilde{\theta}_i; \tilde{\theta}_i)$, giving $\min_\theta L_{DQN}(\theta; \tilde{\theta}_i) \le L_{MSBE}(\tilde{\theta}_i)$.

When DQN diverges, $\min_\theta L_{DQN}(\theta; \tilde{\theta}_i)$ diverges with increasing $i$, and therefore $L_{MSBE}(\tilde{\theta}_i)$ must also diverge. Therefore at each iteration, while the minimizer $\tilde{\theta}_{i+1}$ minimizes the $i$-th DQN loss $L_{DQN}(\theta; \tilde{\theta}_i)$, it can increase the upper bound of the $(i+1)$-th DQN loss, i.e. $L_{MSBE}(\tilde{\theta}_{i+1})$. We want both $L_{DQN}$ and $L_{MSBE}$ to decrease in learning and we define the convergent DQN (C-DQN) loss as

$$L_{CDQN}(\theta; \tilde{\theta}_i) := \mathbb{E}\left[\max\left\{\ell_{DQN}(\theta; \tilde{\theta}_i), \ell_{MSBE}(\theta)\right\}\right], \qquad (12)$$

where $\ell_{MSBE}(\theta) := (Q_\theta(s_t, a_t) - r_t - \gamma \max_{a'} Q_\theta(s_{t+1}, a'))^2$ and $L_{MSBE} = \mathbb{E}[\ell_{MSBE}]$.

**Theorem 2.** *The C-DQN loss satisfies $\min_\theta L_{CDQN}(\theta; \tilde{\theta}_{i+1}) \le \min_\theta L_{CDQN}(\theta; \tilde{\theta}_i)$, given $\tilde{\theta}_{i+1} = \arg\min_\theta L_{CDQN}(\theta; \tilde{\theta}_i)$.*

We have

$$\min_\theta L_{CDQN}(\theta; \tilde{\theta}_{i+1}) \le L_{CDQN}(\tilde{\theta}_{i+1}; \tilde{\theta}_{i+1}) = L_{MSBE}(\tilde{\theta}_{i+1}), \qquad (13)$$

$$L_{MSBE}(\tilde{\theta}_{i+1}) \le \mathbb{E}\left[\max\left\{\ell_{DQN}(\tilde{\theta}_{i+1}; \tilde{\theta}_i), \ell_{MSBE}(\tilde{\theta}_{i+1})\right\}\right] = L_{CDQN}(\tilde{\theta}_{i+1}; \tilde{\theta}_i) \le \min_\theta L_{CDQN}(\theta; \tilde{\theta}_i). \qquad (14)$$

Therefore, we obtain the desired non-increasing condition $\min_\theta L_{CDQN}(\theta; \tilde{\theta}_{i+1}) \le \min_\theta L_{CDQN}(\theta; \tilde{\theta}_i)$, which means that the iteration $\tilde{\theta} \leftarrow \arg\min_\theta L_{CDQN}(\theta; \tilde{\theta})$ is convergent, in the sense that the loss is bounded from below and non-increasing.

C-DQN as defined above is convergent for a given fixed dataset. Although the analysis starts from the assumption that $\tilde{\theta}_i$ exactly minimizes the loss, in fact, it is not necessary. In practice, at the moment when the target network $\tilde{\theta}$ is updated by $\theta$, the loss immediately becomes equal to $L_{MSBE}(\theta)$ which is bounded from above by the loss $L_{CDQN}(\theta; \tilde{\theta})$ before the target network update. Therefore, as long as the loss is consistently optimized during the optimization process, the non-increasing property of the loss holds throughout training. We find that it suffices to simply replace the loss used in DQN by Eq. (12) to implement C-DQN. As we empirically find that $L_{MSBE}$ in RG is always much smaller than $L_{DQN}$, we expect C-DQN to put more emphasis on the DQN loss and to have learning behaviour similar to DQN. C-DQN can also be augmented by various extensions of DQN, such as double Q-learning, distributional DQN and soft Q-learning (Van Hasselt et al., 2016; Bellemare et al., 2017; Haarnoja et al., 2017), by modifying the losses $\ell_{DQN}$ and $\ell_{MSBE}$ accordingly. The mean squared loss can also be replaced by the Huber loss (smooth $\ell 1$ loss) as commonly used in DQN implementations. More discussions on the properties of C-DQN are provided in the appendix.

## 5 Experiments

### 5.1 Comparison of C-DQN, DQN and RG

We focus on the *Atari 2600* benchmark as in Mnih et al. (2015), and use the dueling network architecture and prioritized sampling, with double Q-learning where applicable (Wang et al., 2016;

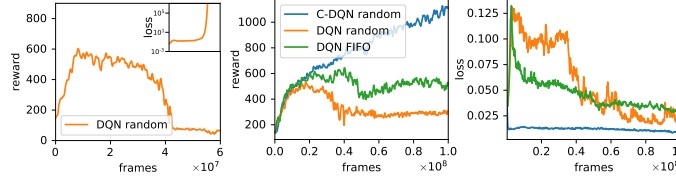

Figure 4: Training performance and training loss on games *Pong* (left) and *Space Invaders* (right).

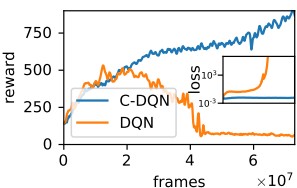

Figure 5: Training performance and loss on *Space Invaders* when half of the data are randomly discarded.

Figure 6: Training performance and training loss on *Space Invaders* when the memory adopts a random replacement strategy (left) and when the memory is smaller and adopts different strategies (middle and right).

Schaul et al., 2015; Van Hasselt et al., 2016). We refer to the combination of the original DQN and these techniques as DQN, and similarly for C-DQN and RG. Details of experimental settings are given in the appendix and our codes are available in the supplementary material.

As C-DQN, DQN and RG only differ in their loss functions, we follow the hyperparameter settings in Hessel et al. (2018) for all the three algorithms and compare them on two well-known games, *Pong* and *Space Invaders*, and the learning curves for performance and loss are shown in Fig. 4. We see that both C-DQN and DQN can learn the tasks, while RG almost does not learn, despite that RG has a much smaller loss. This coincides with our prediction in Sec. 3, which explains why there are very few examples of successful applications of RG to realistic problems. The results show that C-DQN as a convergent method indeed performs well in practice and has performance comparable to DQN for standard tasks. Results for a few other games are given in the appendix.

## 5.2 LEARNING FROM INCOMPLETE TRAJECTORIES OF EXPERIENCE

To give an example in which DQN is prone to diverge, we consider learning from incomplete trajectories of experience, i.e. given a transition $(s_t, a_t, s_{t+1})$ in the dataset, the subsequent transition $(s_{t+1}, a_{t+1}, s_{t+2})$ may be absent from the dataset. This makes DQN prone to diverge because while DQN learns $Q_\theta(s_t, a_t)$ based on $\max_{a'} Q_\theta(s_{t+1}, a')$, there is a possibility that $\max_{a'} Q_\theta(s_{t+1}, a')$ has to be inferred and cannot be directly learned from any data. To create such a setting, we randomly discard half of the transition data collected by the agent and keep the other experimental settings unchanged, except that the mean squared error is used instead of the Huber loss to allow for divergence in gradient. We find that whether or not DQN diverges or not is task-dependent in general, with a larger probability to diverge for more difficult tasks, and the result for *Space Invaders* is shown in Fig. 5. We see that while DQN diverges, C-DQN learns stably and the speed of its learning is only slightly reduced. This confirms that C-DQN is convergent regardless of the structure of the learned data, and implies that C-DQN may be potentially more suitable for offline learning and learning from observation when compared with DQN.

A similar situation arises when one does not use the first-in-first-out (FIFO) strategy to replace old data in the replay memory (i.e. the dataset) with new data when the dataset is full, but replaces old data randomly with new data. In Fig. 6, we show that conventional DQN can actually diverge in this simple setting. In the existing DQN literature, this replacement strategy is often ignored, while here it can been seen to be an important detail that affects the results, and in practice, FIFO is almost always used. However, FIFO makes the memory data less diverse and less informative, and it increases the possibility of the oscillation of the co-evolvement of the policy and the replay memory, and as a result, a large size of the replay memory is often necessary for learning. In Fig. 6, we show that when the size of the replay memory is reduced by a factor of 10, C-DQN can benefit from utilizing the random replacement strategy while DQN cannot, and C-DQN can reach a higher performance. Note that DQN does not diverge in this case, probably because the replay memory is

Figure 7: Training performance on several difficult games in *Atari 2600*, with learning rate $4 \times 10^{-5}$. Each line represents a single run and the shaded regions show the standard deviation. The discount factors are shown in the titles and all DQN agents have significant instabilities or divergence in loss.

less off-policy when it is small, which alleviates divergence. The result opens up a new possibility of RL of only storing and learning important data to improve efficiency, which cannot be realized stably with DQN but is possible with C-DQN.

### 5.3 Difficult games in Atari 2600

In this section we consider difficult games in *Atari 2600*. While DQN often becomes unstable when the discount factor $\gamma$ gets increasingly close to 1, in principle, C-DQN can work with any $\gamma$. However, we find that a large $\gamma$ does not always result in better performance in practice, because a large $\gamma$ requires the agent to learn to predict rewards that are far in the future, which are often irrelevant for learning the task. We also notice that when $\gamma$ is larger than 0.9999, the order of magnitude of $(1 - \gamma)Q_\theta$ gets close to the intrinsic noise caused by the finite learning rate and learning can stagnate. Therefore, we require $\gamma$ to satisfy $0.99 \leq \gamma \leq 0.9998$, and use a simple heuristic algorithm to evaluate how frequent reward signals appear so as to determine $\gamma$ for each task, which is discussed in detail in the appendix. We also take this opportunity to evaluate the mean $\mu$ of $Q$ and the scale $\sigma$ of the reward signal using sampled trajectories, and make our agent learn the normalized value $\frac{Q-\mu}{\sigma}$ instead of the original $Q$. We do not clip the reward and follow Pohlen et al. (2018) to make the neural network learn a transformed function which squashes the Q function approximately by the square root.

With C-DQN and large $\gamma$ values, several difficult tasks which previously could not be solved by simple DQN variants can now be solved, as shown in Fig. 7. We find that especially for *Skiing*, *Private Eye* and *Venture*, the agent significantly benefits from large $\gamma$ and achieves a higher best performance in training, even though *Private Eye* and *Venture* are partially observable tasks and not fully learnable, which leads to unstable training performance. Evaluation of the test performance and details of the settings are given in the appendix. Notably, we find that C-DQN achieves the state-of-the-art test performance on *Skiing* despite the simplicity of the algorithm.

## 6 Conclusion and future perspectives

We have discussed the inefficiency issues regarding RG and gradient-TD methods, and addressed the long-standing problem of convergence in Q-learning by proposing a convergent DQN algorithm, and we have demonstrated the effectiveness of C-DQN on the *Atari 2600* benchmark. With the stability of C-DQN, we can now consider the possibility of tuning $\gamma$ freely without sacrificing stability, and consider the possibility of learning only important state transitions to improve efficiency. C-DQN can be applied to difficult tasks for which DQN suffers from instability. It may also be combined with other strategies that involve target networks and potentially improve their stability.

There are many outstanding issues concerning C-DQN. The loss used in C-DQN is non-smooth, and it is not clear how this affects the optimization and the learning dynamics. In our experiments this does not appear to be a problem, but deserves further investigation. When the transitions are stochastic, the loss $L_{MSBE}$ used in C-DQN does not converge exactly to the solution of the Bellman equation, and therefore it would be desirable if C-DQN can be improved so that stochastic transitions can be learned without bias. It would be interesting to investigate how it interplays with DQN extensions such as distributional DQN and soft Q-learning, and it is not clear whether the target network in C-DQN can be updated smoothly as in Lillicrap et al. (2015).

REPRODUCIBILITY STATEMENT

All our experimental results can be reproduced exactly by our codes provided in the supplementary material, where the scripts and commands are organised according to the section numbers. We also present and discuss our implementation details in the appendix so that one can reproduce our results without referring to the codes.

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

# A   CONVERGENCE OF C-DQN IN STOCHASTIC SETTINGS

When the transition of states is stochastic, it is well-known that the minimum of $L_{MSBE}$ does not exactly correspond to the solution of the Bellman equation, because we have

$$L_{MSBE}(\theta) = \mathbb{E}_{s_{t+1}} \left[ \left( Q_\theta(s_t, a_t) - r_t - \gamma \max_{a'} Q_\theta(s_{t+1}, a') \right)^2 \right]$$

$$= \left( Q_\theta(s_t, a_t) - r_t - \gamma \mathbb{E}_{s_{t+1}} \left[ \max_{a'} Q_\theta(s_{t+1}, a') \right] \right)^2 + \gamma^2 \mathrm{Var}_{s_{t+1}}(\max_{a'} Q_\theta(s_{t+1}, a')),$$
(15)

where $\mathrm{Var}(\cdot)$ represents the variance, and it can be seen that only the first term on the last line corresponds to the solution of the Bellman equation. Because C-DQN involves $L_{MSBE}$, if the underlying task is stochastic, C-DQN may not converge to the optimal solution due to the bias in $L_{MSBE}$. In fact, both the minimum of $L_{MSBE}$ and the solution of the Bellman equation are stationary points for C-DQN. To show this, we first assume that the minimum of $L_{DQN}$ and that of $L_{MSBE}$ are unique and different. If parameters $\theta$ and $\tilde{\theta}_i$ satisfy $\theta = \tilde{\theta}_i = \arg\min L_{DQN}(\cdot; \tilde{\theta}_i)$, i.e., if they are at the converging point of DQN, then we consider an infinitesimal change $\delta\theta$ of $\theta$ following the gradient of $L_{MSBE}$ in an attempt to reduce $L_{MSBE}$. Because we have $L_{MSBE}(\theta) = L_{DQN}(\theta; \theta) = L_{DQN}(\theta; \tilde{\theta}_i)$, as $\theta + \delta\theta$ moves away from the minimum $L_{DQN}(\theta; \tilde{\theta}_i)$, we have

$$L_{DQN}(\theta + \delta\theta; \tilde{\theta}_i) > L_{DQN}(\theta; \tilde{\theta}_i) = L_{MSBE}(\theta) > L_{MSBE}(\theta + \delta\theta),$$
(16)

and therefore for $\theta + \delta\theta$, $L_{DQN}$ is larger than $L_{MSBE}$, and C-DQN will choose to optimize $L_{DQN}$ instead of $L_{MSBE}$ and the parameter will return to the minimum $\theta$, which is the converging point of DQN. On the other hand, given $\theta = \tilde{\theta}_i = \arg\min L_{MSBE}(\cdot)$, if we change $\theta$ by an infinitesimal amount $\delta\theta$ in an attempt to reduce $L_{DQN}(\theta; \tilde{\theta}_i)$, we similarly have

$$L_{MSBE}(\theta + \delta\theta) > L_{MSBE}(\theta) = L_{DQN}(\theta; \tilde{\theta}_i) > L_{DQN}(\theta + \delta\theta; \tilde{\theta}_i),$$
(17)

and therefore, for the same reason C-DQN will choose to optimize $L_{MSBE}$ and the parameter will return to $\theta$. Therefore, C-DQN can converge to both the converging points of DQN and RG. More generally, C-DQN may converge somewhere between the converging points of DQN and RG. It tries to minimize both the loss functions simultaneously, and it stops if this goal cannot be achieved, i.e., if a decrease of one loss increases the other. Interestingly, this does not seem to be a severe problem as demonstrated by the successful application of C-DQN to the *Atari 2600* benchmark (see Sec. 5), because the tasks include a large amount of noise-like behaviour and subtitles such as partially observable states.

## A.1   A CASE STUDY: THE WET-CHICKEN BENCHMARK

To investigate the behaviour of C-DQN more closely, we consider a stochastic toy problem, which is known as the *wet-chicken* benchmark (Tresp, 1994; Hans & Udluft, 2009; 2011). In the problem, a canoeist paddles on a river starting at position $x = 0$, and there is a waterfall at position $x = l = 20$, and the goal of the canoeist is to get as close as possible to the waterfall without reaching it. The canoeist can choose to paddle back, hold the position, or drift forward, which corresponds to a change of -1, 0, or +1 in his/her position $x$, and there is random turbulence $z \sim Uniform(-2.5, +2.5)$, a uniformly distributed random number, that also contributes to the change in $x$ and stochastically perturbs $x$ at each step. The reward is equal to the position $x$, and $x$ is reset back to 0 if he/she reaches the waterfall at $x = 20$. The task does not involve the end of an episode, and the performance is evaluated as the average reward per step. This task is known to be highly stochastic, because the effect of the stochastic perturbation is often stronger than the effect of the action of the agent, and the stochasticity can lead to states that have dramatically different Q function values.

To learn this task, we generate a dataset of 20000 transitions using random actions, and we train a neural network on this dataset using the DQN, the C-DQN and the RG algorithms to learn the Q values. The results are shown in Fig. 8. In the left panel of Fig. 8, it can be seen that while RG significantly underperforms DQN, the performance of C-DQN lies between DQN and RG and is only slightly worse than DQN. This shows that when the task is highly stochastic, although C-DQN

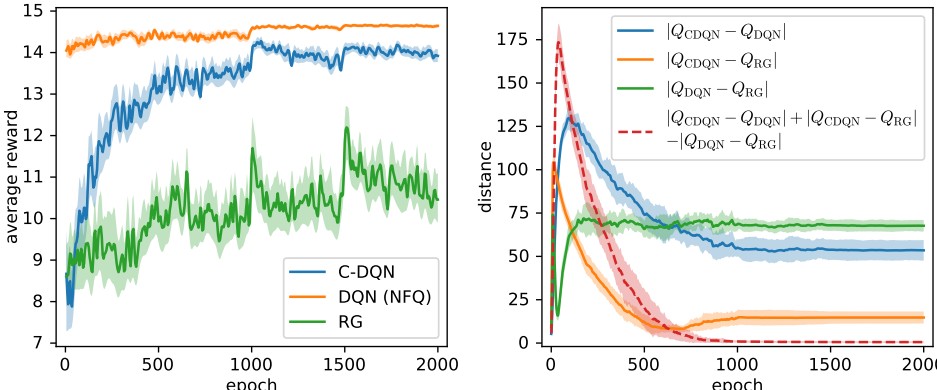

Figure 8: Performance on the wet-chicken benchmark training on a dataset generated by the random policy (left) and the distances among the learned Q functions (right). The experiment is repeated for 10 times, and the standard error of the performance and the standard deviation of the distances are shown as the shaded regions.

may not reach the optimal solution as DQN can do, C-DQN still behaves robustly and produces reasonably satisfactory results, while RG fails dramatically.

To obtain further details of the learned Q functions, we estimate the distance between the Q functions. The distance $|Q_1 - Q_2|$ between two Q functions $Q_1$ and $Q_2$ is estimated as $\sqrt{\sum_{(x,a)}(Q_1(x,a) - Q_2(x,a))^2}$, where the summation on the position $x$ is taken over the discrete set $\{0, 1, 2, ...19\}$. In the right panel of Fig. 8, we show the estimated distances among the learned Q functions of DQN, C-DQN and RG. We see that the distance between $Q_{DQN}$ and $Q_{CDQN}$ increases rapidly in the beginning and then slowly decreases, implying that DQN learns quickly at the beginning, and C-DQN catches up later and reduces its distance to DQN. Notably, we find that the value $|Q_{CDQN} - Q_{DQN}| + |Q_{CDQN} - Q_{RG}| - |Q_{DQN} - Q_{RG}|$ always converges to zero, indicating that the solution found by C-DQN, i.e. $Q_{CDQN}$, lies exactly on the line from $Q_{DQN}$ to $Q_{RG}$, which is consistent with our argument that C-DQN converges somewhere between the converging points of DQN and RG.

Concerning the experimental details, the neural network includes 4 hidden layers, each of which has 128 hidden units and uses the ReLU as the activation function, and the network is optimized using the Adam optimizer (Kingma & Ba, 2014) with default hyperparameters. We use the batch size of 200, and the target network is updated after each epoch, which makes the DQN algorithm essentially the same as the neural fitted Q (NFQ) iteration algorithm (Riedmiller, 2005). The training includes 2000 epochs, and the learning rate is reduced by a factor of 10 and 100 at the 1000th and the 1500th epochs. The discount factor $\gamma$ is set to be 0.97. The position $x$ and the reward are normalized by 20 before training, and the evaluation of the performance is done every 5 epochs, using 300 time steps and repeated for 200 trials. The entire experiment is repeated for 10 times including the data generation process.

## B  ADDITIONAL EXPERIMENTAL RESULTS

In this section we present additional experimental results. In Sec. B.1, we present the results of applying C-DQN to the problem of measurement feedback cooling of quantum quartic oscillators in Wang et al. (2020), where we show that the final performances are more consistent regarding different random seeds and have a smaller variance compared with the results of DQN. Concerning the *Atari 2600* benchmark, in Sec. B.2, results for several other games are presented. In Sec. B.3 we show that C-DQN allows for more flexible update periods of the target network. In Sec. B.4 we report the test performance of C-DQN on the difficult *Atari 2600* games shown in Sec. 5.3, and in Sec. B.5 we discuss the results of C-DQN on the game *Skiing*.

### B.1 RESULTS ON MEASUREMENT FEEDBACK COOLING OF A QUANTUM QUARTIC OSCILLATOR

To show the stability of C-DQN compared with DQN for problems with practical significance, we reproduce the results in Wang et al. (2020), which trains a RL controller to do measurement feedback cooling of a one-dimensional quantum quartic oscillator in numerical simulation. Details of the problem setting are given in Wang et al. (2020) and in our supplementary codes. The training performances for C-DQN and DQN are shown in Fig. 9, where each curve represents a different experiment with a different random seed. As shown in Fig. 9, different C-DQN experiments have similar learning curves and final performances; however, in sharp contrast, those of the DQN experiment have apparently different fluctuating learning curves and the performances are unstable, and some of the repetitions cannot reach a final performance that is comparable to the best-performing ones. The results show that compared with DQN, the outcome of the training procedure of C-DQN is highly stable and reproducible, which can greatly benefit practical applications.

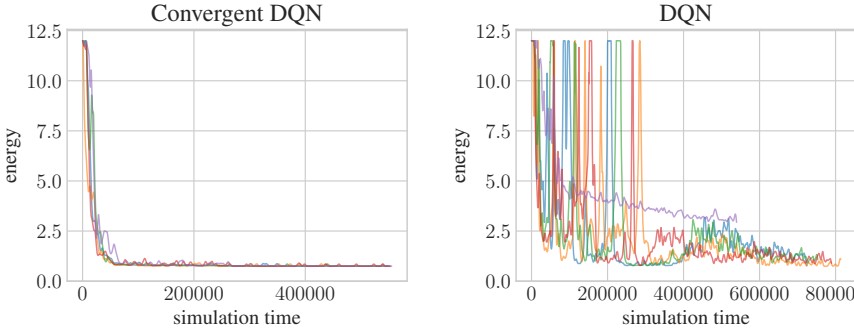

Figure 9: Training performance of C-DQN and DQN on the task of measurement feedback cooling of quartic oscillators. The vertical axis shows the energy of the cooled quartic oscillator, and a smaller energy represents better performance. The horizontal axis shows the simulated time of the oscillator system that is used to train the agent. Each curve represents a separate trial of the experiment.

### B.2 EXPERIMENTAL RESULTS ON OTHER GAMES

In addition to the results in Sec. 5.1, we present results on 6 other *Atari 2600* games comparing C-DQN and DQN, which are shown in Fig. 10.

In general, we find that the loss value of C-DQN is almost always smaller than DQN, and for relatively simple tasks, C-DQN has performance comparable to DQN, but for more difficult tasks, they show different performances with relatively large variance. Specifically, we find that C-DQN has better performance for tasks that require more precise learning and control such as *Atlantis*, and for tasks that are unstable and irregular such as *Video Pinball*. However, for tasks that are highly stochastic and partially observable such as *Fishing Derby* and *Time Pilot*, C-DQN may perform less well compared with DQN, probably because the term $L_{MSBE}$ in $L_{CDQN}$ does not properly account for stochastic transitions.

### B.3 MORE FLEXIBLE UPDATE PERIODS FOR THE TARGET NETWORK

As C-DQN has a better convergence property, it allows for shorter update periods of the target network. Specifically, convergence of C-DQN is obtained as long as the loss decreases during the optimization process after an update of the target network. One period of the update of the target network consists of $N_{\tilde{\theta}}$ iterations of gradient descent on $\theta$ minimizing $L_{CDQN}(\theta; \tilde{\theta}_i)$ or $L_{DQN}(\theta; \tilde{\theta}_i)$ with the target network $\tilde{\theta}_i$, and then using $\theta$ as the next target network $\tilde{\theta}_{i+1}$, where $N_{\tilde{\theta}}$ represents the update period of the target network. In previous works on DQN, $N_{\tilde{\theta}}$ is set to be 2000 or 2500 (Hessel et al., 2018; Mnih et al., 2015), and DQN may experience instability for a too small $N_{\tilde{\theta}}$. However, we empirically find that for many tasks in *Atari 2600*, $N_{\tilde{\theta}}$ can be reduced to 200 or even 20 without instability for C-DQN. Therefore, we find that C-DQN requires less fine tuning on the

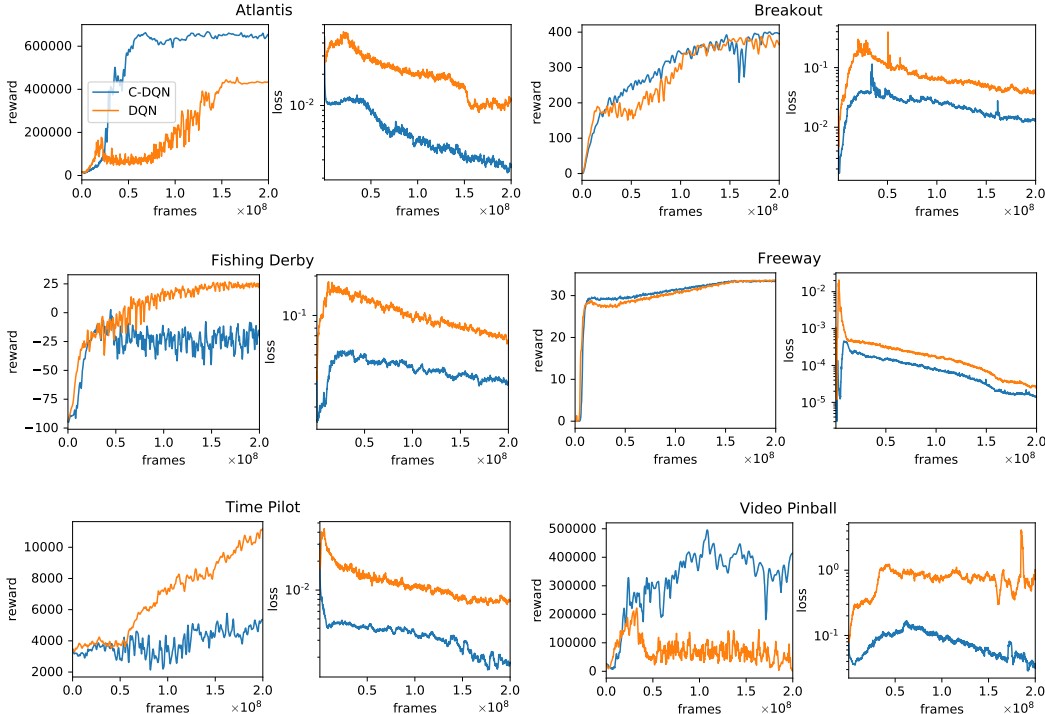

Figure 10: Training performance and training loss of C-DQN and DQN on several other *Atari 2600* games, using the same experimental settings as in Sec. 5.1.

hyperparameter $N_{\tilde{\theta}}$ compared with DQN. The experimental results on *Space Invaders* and *Hero* are shown in Fig. 11, using the experimental settings in Sec. 5.1. We see that C-DQN has a generally higher performance compared to DQN when $N_{\tilde{\theta}}$ becomes small, and the performance of DQN is sometimes unstable and is sensitive to the value of $N_{\tilde{\theta}}$.

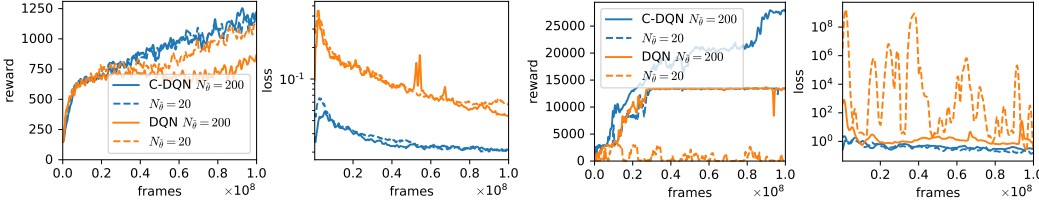

Figure 11: Training performance using different update periods of the target network on games *Space Invaders* (left) and *Hero* (right). In the game *Hero*, there appears to be a local optimum with reward 13000 where the learning can fail to make progress, which is also seen in Fig. 15.

## B.4 TEST PERFORMANCE ON DIFFICULT ATARI GAMES

In this section we report the test performance of the agents in Sec. 5.3 and compare with existing works. As the learned policy of the agents has large fluctuations in performance due to noise, local optima, and insufficient learning when the task is difficult, instead of using the agent at the end of training, we use the best-performing agent during training to evaluate the test performance. Specifically, we save a checkpoint of the parameter $\theta$ of the agent every $10^4$ steps, and we choose the three best-performing agents by comparing their training performances, computed as the average of the 40 nearby episodes around each of the checkpoints. After the training, we carry out a separate validation process using 400 episodes to find the best-performing one among the three agents, and then, we evaluate the test performance of the validated best agent by another 400 episodes, using a different random seed. The policy during evaluation is the $\epsilon$-greedy policy with $\epsilon = 0.01$,, with no-

Table 1: Test performance on difficult *Atari 2600* games, corresponding to the results in Sec. 5.3 in the main text, evaluated using no-op starts and without sticky actions (Machado et al., 2018). The DQN results are produced using the same experimental settings as the C-DQN experiments except for the loss function. Human results and results for Agent57 are due to Badia et al. (2020), and results for Rainbow DQN are due to Hessel et al. (2018). The human results only represent the performance of an average person, not a human expert, and the human results correspond to reasonably adequate performance instead of the highest possible performance of human.

| Task | C-DQN | DQN | Human | Rainbow DQN | Agent57 (SOTA) |
|---|---|---|---|---|---|
| Skiing | **-3697 ± 157** | -29751 ± 224 | -4337 | -12958 | -4203 ± 608 |
| Tennis | 10.9 ± 6.3 | -2.6 ± 1.4 | -8.3 | 0.0 | 23.8 ± 0.1 |
| Private Eye | 14730 ± 37 | 7948 ± 749 | 69571 | 4234 | 79716 ± 29545 |
| Venture | 893 ± 51 | 386 ± 85 | 1188 | 5.5 | 2624 ± 442 |

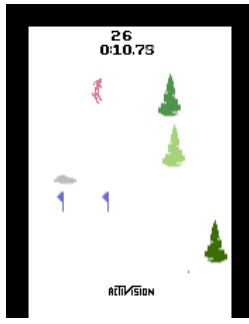

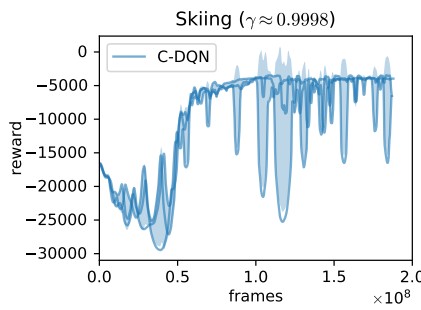

Figure 12: A screenshot of the game *Skiing* in *Atari 2600*.

Figure 13: Training performance of C-DQN on *Skiing* with learning rate $2 \times 10^{-5}$, following the experimental procedure in Sec. 5.3. The standard deviation among the three runs are shown as the shaded region.

op starts[5] (Mnih et al., 2015). The average of the test performances of the 3 runs in our experiments are shown in Table 1 together with the standard error, compared with existing works and the human performance.

As we have basically followed the conventional way of training DQN on *Atari 2600* as in Mnih et al. (2015) and Hessel et al. (2018), our C-DQN and the Rainbow DQN in Hessel et al. (2018) allow for a fair comparison because they use the same amount of computational budget and a similar neural network architecture.[6] In Table 1, we see that in these four difficult *Atari 2600* games Rainbow DQN fails to make progress in learning, and C-DQN can achieve performances higher than Rainbow DQN and show non-trivial learning behaviour. The results of Agent57 is for reference only, which represents the currently known best performance on *Atari 2600* in general and does not allow for a fair comparison with C-DQN, as it involves considerably more computation, more sophisticated methods and larger neural networks. We find that our result on the game *Skiing* is exceptional, which is discussed in the next section.

## B.5 THE ATARI GAME *Skiing*

In Table 1, one exceptional result is that C-DQN achieves a performance higher than Agent57 on the game *Skiing*, actually, using an amount of computation that is less than $0.1\%$ of that of Agent57. We find that this performance is higher than any other known result so far and thus achieves the state-of-the-art (SOTA) for this specific game. To elucidate the underlying reason, we describe this game first.

---

[5]No-op starts mean that at the start of each episode the *no-operation* action is executed randomly for 1 to 30 frames.

[6]In fact, a fair comparison with Rainbow DQN can be made except for the case of *Skiing*, because reward clipping adopted by Rainbow DQN does not permit the learning of *Skiing*. Nevertheless, this does not affect our conclusion.

A screenshot of the game *Skiing* is shown in Fig. 12. This game is similar to a racing game. The player is required to go downhill and reach the goal as fast as possible, and the time elapsed before reaching the goal is the minus reward. At each time step, the player receives a small minus reward which represents the accumulated time, until the goal is reached and the game ends. In addition, the player is required to pass through the gates along his/her way, which are represented by the two small flags shown in Fig. 12, and whenever the player fails to pass a gate, a 5-second penalty is added to the elapsed time when the player reaches the final goal. The number of gates that have not been passed are shown at the top of the game screen. Using the standard setting in Mnih et al. (2015), the number of state transitions for an episode of this game is $\sim 1300$ for the random policy, $\sim 4500$ when the player slows down significantly, and $\sim 500$ when the policy is near-optimal.

Since the penalty for not passing a gate is given only at the end of the game, the agent needs to relate the penalty at the end of the game to the events that happen early in the game, and therefore the discount factor $\gamma$ should be at least around $1 - \frac{1}{500}$ to make learning effective. However, the learning may still stagnate if $\gamma$ is not larger, because when $\gamma$ is small, the agent prefers taking a longer time before reaching the goal, so that the penalty at the end is delayed and the Q function for the states in the early game is increased, which will increase the episode length and make a larger $\gamma$ necessary. Therefore, we have chosen to tune our hyperparameter setting so that $\gamma \approx 1 - \frac{1}{5000}$ is obtained on this game (see Sec. E.3), and we find that our C-DQN agent successfully learns with the $\gamma$ and produces a new record on this game.[7] The large fluctuations in its training performance shown in Sec. 5.3 are mostly due to the noise coming from the finite learning rate, which can be confirmed by repeating the experiments with a smaller learning rate, shown in Fig. 13. However, in this case the learning easily gets trapped in local optima and the final test performance is worse. Note that in fact, we cannot fairly compare our result with Agent57, because we have tuned our hyperparameters so that the obtained $\gamma$ is in favour of this game, while Agent57 uses a more general bandit algorithm to adaptively determine $\gamma$.

## C  RELATED WORKS

In this section, we present some related works for further references. The slow convergence of RG compared with TD methods has been shown in Schoknecht & Merke (2003). The $O(N^2)$ scaling property can also be derived by considering specific examples of Markov chains, such as the "Hall" problem as pointed out in Baird (1995). The convergence property of RG-like algorithms has been analysed in Antos et al. (2008), assuming that the underlying Markov process is $\beta$-mixing, i.e. it converges to a stable distribution exponentially fast. However, this assumption is often impractical, which may have underlain the discrepancy between the theoretical results of RG and the experimental effectiveness. There is an improved version of RG proposed in Zhang et al. (2020), and RG has been applied to robotics in Johannink et al. (2019). Concerning DQN, there have been many attempts to stabilize the learning, to remove the target network, and to use a larger $\gamma$. Pohlen et al. (2018) introduces a temporal consistency loss to reduce the difference between $Q_\theta$ and $Q_{\bar{\theta}}$ on $s_{t+1}$, and the authors showed that the resulting algorithm can learn with $\gamma = 0.999$. A variant of it is proposed in Ohnishi et al. (2019). Kim et al. (2019) and Bhatt et al. (2019) propose extensions for DQN and show that the resulting DQN variants can sometimes operate without a target network when properly tuned. Achiam et al. (2019) gives an analysis of the divergence and proposes to use a method similar to natural gradient descent to stabilize Q-learning; however, it is computationally heavy as it uses second-order information, and therefore it cannot be used efficiently with large neural networks. Recently, the strategy of using target networks in DQN has been shown to be useful for TD learning as well by Zhang et al. (2021). Some other works have been discussed in the main text and we do not repeat them here.

## D  CALCULATION DETAILS OF THE CONDITION NUMBER

In this section we provide the calculation details of Sec. 3.1. Given the loss function

$$L = \frac{1}{N} \left[ \sum_{t=0}^{N-2} (Q_t - r_t - Q_{t+1})^2 + (Q_{N-1} - r_{N-1})^2 \right], \tag{18}$$

---

[7]The single highest performance we observed was around $-3350$, and the optimal performance in this game is reported to be $-3272$ in Badia et al. (2020).

where $Q(s_t, a_t)$ is denoted by $Q_t$, we add an additional term $(Q_0)^2$ to it, and the loss function becomes

$$
\begin{aligned}
L &= \frac{1}{N}[Q_0^2 + \sum_{t=0}^{N-2}(Q_t - r_t - Q_{t+1})^2 + (Q_{N-1} - r_{N-1})^2] \\
&= \frac{1}{N}[Q_0^2 + \sum_{t=0}^{N-2}\left(Q_t^2 - 2r_tQ_t + 2r_tQ_{t+1} - 2Q_tQ_{t+1} + r_t^2 + Q_{t+1}^2\right) \\
&\quad + \left(Q_{N-1}^2 - 2r_{N-1}Q_{N-1} + 2r_{N-1}^2\right)].
\end{aligned}
\tag{19}
$$

To calculate the condition number of the Hessian matrix, we ignore the prefactor $\frac{1}{N}$ and only evaluate the second-order derivatives. From Eq. (19), it is straightforward to obtain

$$
\begin{aligned}
\frac{\partial^2 L}{\partial Q_t^2} &= 4, \qquad t \in \{0, 1, ...N-1\} \\
\frac{\partial^2 L}{\partial Q_t \partial Q_{t+1}} &= -2, \qquad t \in \{0, 1, ...N-2\}
\end{aligned}
\tag{20}
$$

and therefore the Hessian matrix is

$$
\begin{pmatrix}
4 & -2 & & \\
-2 & 4 & \ddots & \\
& \ddots & \ddots & -2 \\
& & -2 & 4
\end{pmatrix}_{N \times N}.
\tag{21}
$$

The eigenvectors of this matrix can be explicitly obtained due to the special structure of the matrix, which are given by $(\sin\frac{k\pi}{N+1}, \sin\frac{2k\pi}{N+1}, ... \sin\frac{Nk\pi}{N+1})^T$ for $k \in \{1, 2, ...N\}$, and the corresponding eigenvalues are given by $4 - 4\cos\frac{k\pi}{N+1}$. To show this, we first simplify the Hessian matrix by removing its constant diagonal 4, which only contributes to a constant 4 in the eigenvalues, and then we multiply the eigenvectors by the eigenvalues

$$
\left(\sin\frac{k\pi}{N+1}, \sin\frac{2k\pi}{N+1}, ... \sin\frac{Nk\pi}{N+1}\right)^T \cdot (-4\cos\frac{k\pi}{N+1}) =
\tag{22}
$$

$$
-2\left((\sin\frac{0 \cdot k\pi}{N+1} + \sin\frac{2k\pi}{N+1}), (\sin\frac{k\pi}{N+1} + \sin\frac{3k\pi}{N+1}), ...(\sin\frac{(N-1)k\pi}{N+1} + \sin\frac{(N+1)k\pi}{N+1})\right)^T
\tag{23}
$$

due to $\sin\frac{nk\pi}{N+1}\cos\frac{k\pi}{N+1} = \frac{1}{2}\left(\sin\frac{(n-1)k\pi}{N+1} + \sin\frac{(n+1)k\pi}{N+1}\right)$. As we also have $\sin\frac{0 \cdot k\pi}{N+1} = \sin\frac{(N+1)k\pi}{N+1}) = 0$, the product of the vector and the eigenvalue is exactly equal to the product of the vector and the Hessian matrix. As the vectors $(\sin\frac{k\pi}{N+1}, \sin\frac{2k\pi}{N+1}, ... \sin\frac{Nk\pi}{N+1})^T$ are linearly independent for $k \in \{1, 2, ...N\}$, they are all the eigenvectors of the Hessian matrix, and therefore the eigenvalues are $4 - 4\cos\frac{k\pi}{N+1}$. The condition number is then $\kappa = \frac{4-4\cos\frac{N\pi}{N+1}}{4-4\cos\frac{\pi}{N+1}}$. Using the Taylor series expansion, the denominator in the expression of $\kappa$ is $4 - 4\cos\frac{\pi}{N+1} = \frac{2\pi^2}{(N+1)^2} + O(\frac{1}{N^4})$, and therefore for large $N$, $\kappa \approx \frac{4-4\cos\pi}{4-4\cos\frac{\pi}{N+1}} \sim O(N^2)$.

When the state $s_{N-1}$ makes a transition to $s_0$, with the discount factor $\gamma$, the loss is given by

$$
L = \frac{1}{N}\left[\sum_{t=0}^{N-2}(Q_t - r_t - \gamma Q_{t+1})^2 + (Q_{N-1} - r_{N-1} - \gamma Q_0)^2\right].
\tag{24}
$$

Using the same calculation, the non-zero second-order derivatives are given by

$$
\begin{aligned}
\frac{\partial^2 L}{\partial Q_t^2} &= 2 + 2\gamma^2, \qquad t \in \{0, 1, ...N-1\} \\
\frac{\partial^2 L}{\partial Q_t \partial Q_{t+1}} &= -2\gamma, \qquad t \in \{0, 1, ...N-1\}, \qquad Q_N = Q_0,
\end{aligned}
\tag{25}
$$

and the Hessian matrix is cyclic. Assuming that $N$ is even, the eigenvectors are given by $(\sin\frac{2k\pi}{N}, \sin\frac{4k\pi}{N}, ... \sin\frac{2Nk\pi}{N})^T$ and $(\cos\frac{2k\pi}{N}, \cos\frac{4k\pi}{N}, ... \cos\frac{2Nk\pi}{N})^T$ for $k \in \{1, 2, ... \frac{N}{2}\}$, with eigenvalues given by $2(1 + \gamma^2) - 4\gamma\cos\frac{2k\pi}{N}$. The result can be similarly confirmed by noticing the relation $\sin\frac{2nk\pi}{N} \cdot (-4\gamma\cos\frac{2k\pi}{N}) = -2\gamma(\sin\frac{2(n-1)k\pi}{N} + \sin\frac{2(n+1)k\pi}{N})$ and the periodicity $\sin\frac{2(N+1)k\pi}{N} = \sin\frac{2k\pi}{N}$, and similarly $\cos\frac{2nk\pi}{N} \cdot (-4\gamma\cos\frac{2k\pi}{N}) = -2\gamma(\cos\frac{2(n-1)k\pi}{N} + \cos\frac{2(n+1)k\pi}{N})$ and $\cos\frac{2(N+1)k\pi}{N} = \cos\frac{2k\pi}{N}$, which proves that they are indeed the eigenvectors and eigenvalues. At the limit of $N \to \infty$, we have $\kappa = \frac{2(1+\gamma^2)-4\gamma\cos\pi}{2(1+\gamma^2)-4\gamma\cos\frac{2\pi}{N}} \approx \frac{2(1+\gamma^2)+4\gamma}{2(1+\gamma^2)-4\gamma} = \frac{(1+\gamma)^2}{(1-\gamma)^2}$. Using $\gamma \approx 1$, we obtain $\kappa \sim O\left(\frac{1}{(1-\gamma)^2}\right)$.

# E   EXPERIMENTAL DETAILS ON ATARI 2600

## E.1   GENERAL SETTINGS

We follow Mnih et al. (2015) to preprocess the frames of the games by taking the maximum of the recent two frames and changing them to the grey scale. However, instead of downscaling them to 84×84 images, we downscale exactly by a factor of 2, which results in 80×105 images as in Ecoffet et al. (2021). This is to preserve the sharpness of the objects in the images and to preserve translational invariance of the objects. For each step of the agent, the agent stacks the frames seen in the recent 4 steps as the current observation, i.e. state $s_t$, and decides an action $a_t$ and executes the action repeatedly for 4 frames in the game and accumulates the rewards during the 4 frames as $r_t$. Thus, the number of frames is 4 times the number of steps of the agent. One iteration of the gradient descent is performed for every 4 steps of the agent, and the agent executes the random policy for 50000 steps to collect some initial data before starting the gradient descent. The replay memory stores 1 million transition data, using the first-in-first-out strategy unless otherwise specified. The neural network architecture is the same as the one in Wang et al. (2016), except that we use additional zero padding of 2 pixels at the edges of the input at the first convolutional layer, so that all pixels in the 80×105 images are connected to the output. Following to Hessel et al. (2018), we set the update period of the target network to be 8000 steps, i.e. 2000 gradient descent iterations, using the Adam optimizer (Kingma & Ba, 2014) with a mini-batch size of 32 and default $\beta_1, \beta_2$ hyperparameters, and we make the agent regard the loss of one life in the game as the end of an episode. The discount factor $\gamma$ is set to be 0.99 unless otherwise specified, and the reward clipping to $[-1, 1]$ is applied except in Sec. 5.3 and E.3.

**Gradient of $L_{CDQN}$ upon updating the target network**   Although we use gradient descent to minimize $L_{CDQN}$, when we update the target network by copying from $\theta$ to $\tilde{\theta}$, $\ell_{DQN}(\theta; \tilde{\theta}_i)$ is exactly equal to $\ell_{MSBE}(\theta)$ and the gradient of $L_{CDQN}$ is undefined. In this case, we find that one may simply use the gradient computed from $\ell_{DQN}$ without any problem, and one may rely on the later gradient descent iterations to reduce the loss $L_{CDQN}$. In our experiments, we further bypass this issue by using the parameters $\theta$ at the previous gradient descent step instead of the current step to update $\tilde{\theta}$, so that $\theta$ does not become exactly equal to $\tilde{\theta}$. This strategy is valid because the consecutive gradient descent steps are supposed to give parameters that minimize the loss almost equally well, and the parameters should have similar values. In our implementation of DQN, we also update the target network in this manner to have a fair comparison with C-DQN.

**Loss**   As mentioned in Sec. 4.2, either the mean squared error or the Huber loss can be used to compute the loss functions $\ell_{DQN}$ and $\ell_{MSBE}$. In Sec. 5.2 we use one half of the mean squared error, and the loss functions are given by

$$\ell_{DQN}(\theta; \tilde{\theta}) = \frac{1}{2}\left(Q_\theta(s_t, a_t) - r_t - \gamma\max_{a'}Q_{\tilde{\theta}}(s_{t+1}, a')\right)^2,$$
$$\ell_{MSBE}(\theta) = \frac{1}{2}\left(Q_\theta(s_t, a_t) - r_t - \gamma\max_{a'}Q_\theta(s_{t+1}, a')\right)^2. \tag{26}$$

In Sec. 5.1 we use the Huber loss, and the loss functions are given by

$$
\ell_{DQN}(\theta; \tilde{\theta}) = \ell_{Huber}\left(Q_\theta(s_t, a_t), \; r_t + \gamma \max_{a'} Q_{\tilde{\theta}}(s_{t+1}, a')\right),
$$

$$
\ell_{MSBE}(\theta) = \ell_{Huber}\left(Q_\theta(s_t, a_t), \; r_t + \gamma \max_{a'} Q_\theta(s_{t+1}, a')\right),
$$

$$
\ell_{Huber}(x, y) = \begin{cases} \frac{1}{2}(x-y)^2, & \text{if } |x-y| < 1, \\ |x-y| - \frac{1}{2}, & \text{if } |x-y| \geq 1. \end{cases}
\tag{27}
$$

In Sec. 5.3, we let the agents learn a normalized Q function $\hat{Q} \equiv \frac{Q-\mu}{\sigma}$, and we use the strategy in Pohlen et al. (2018) to squash the Q function approximately by the square root before learning. The relevant transformation function $\mathcal{T}$ is defined by

$$
\mathcal{T}(\hat{Q}) := \text{sign}(\hat{Q})\left(\sqrt{|\hat{Q}|+1} - 1\right) + \epsilon_\mathcal{T} \hat{Q},
\tag{28}
$$

and its inverse is given by

$$
\mathcal{T}^{-1}(f) = \text{sign}(f)\left(\left(\frac{\sqrt{1 + 4\epsilon_\mathcal{T}\left(|f| + 1 + \epsilon_\mathcal{T}\right)} - 1}{2\epsilon_\mathcal{T}}\right)^2 - 1\right).
\tag{29}
$$

In our experiments we set $\epsilon_\mathcal{T} = 0.01$ as in Pohlen et al. (2018), and the loss functions are

$$
\ell_{DQN}(\theta; \tilde{\theta}) = \ell_{Huber}\left(f_\theta(s_t, a_t), \; \mathcal{T}\left(\hat{r}_t + \gamma \mathcal{T}^{-1}\left(\max_{a'} f_{\tilde{\theta}}(s_{t+1}, a')\right)\right)\right),
$$

$$
\ell_{MSBE}(\theta) = \ell_{Huber}\left(f_\theta(s_t, a_t), \; \mathcal{T}\left(\hat{r}_t + \gamma \mathcal{T}^{-1}\left(\max_{a'} f_\theta(s_{t+1}, a')\right)\right)\right),
\tag{30}
$$

where $f_\theta$ is the neural network, and $\mathcal{T}^{-1}(f_\theta(s_t, a_t))$ represents the learned normalized Q function $\hat{Q}_\theta(s_t, a_t)$, and $\hat{r}_t$ is the reward that is modified together with the normalization of the Q function, which is discussed in Sec. E.3.

When we plot the figures, for consistency, we always report the mean squared errors as the loss functions, which are given by

$$
\ell_{DQN}(\theta; \tilde{\theta}) = \left(Q_\theta(s_t, a_t) - r_t - \gamma \max_{a'} Q_{\tilde{\theta}}(s_{t+1}, a')\right)^2,
$$

$$
\ell_{MSBE}(\theta) = \left(Q_\theta(s_t, a_t) - r_t - \gamma \max_{a'} Q_\theta(s_{t+1}, a')\right)^2,
\tag{31}
$$

or,

$$
\ell_{DQN}(\theta; \tilde{\theta}) = \left(f_\theta(s_t, a_t) - \mathcal{T}\left(\hat{r}_t + \gamma \mathcal{T}^{-1}\left(\max_{a'} f_{\tilde{\theta}}(s_{t+1}, a')\right)\right)\right)^2,
$$

$$
\ell_{MSBE}(\theta) = \left(f_\theta(s_t, a_t) - \mathcal{T}\left(\hat{r}_t + \gamma \mathcal{T}^{-1}\left(\max_{a'} f_\theta(s_{t+1}, a')\right)\right)\right)^2.
\tag{32}
$$

When we use double Q-learning (Van Hasselt et al., 2016) in our experiments, all $\max_{a'} Q_{\tilde{\theta}}(s_{t+1}, a')$ and $\max_{a'} f_{\tilde{\theta}}(s_{t+1}, a')$ terms in the above equations are actually replaced by $Q_{\tilde{\theta}}\left(s_{t+1}, \arg\max_{a'} Q_\theta(s_{t+1}, a')\right)$ and $f_{\tilde{\theta}}\left(s_{t+1}, \arg\max_{a'} f_\theta(s_{t+1}, a')\right)$. This modification do not change the non-increasing property of $L_{CDQN}$, which can be confirmed easily.

**Other details** We find that the exploration can often be insufficient when the game is difficult, and we follow Van Hasselt et al. (2016) and use a slightly more complicated schedule for the $\epsilon$ parameter in the $\epsilon$-greedy policy, slowly annealing $\epsilon$ from 0.1 to 0.01. We have a total computation budget of $5 \times 10^7$ steps for each agent, and at the $j$-th step of the agent, for $j \leq 50000$, we set $\epsilon = 1$ since the initial policy is random; for $50000 > j \geq 10^6$, $\epsilon$ is exponentially annealed to 0.1 following $\epsilon = e^{j/\tau}$, with $\tau = 10^6/\ln(0.1)$; for $10^6 > j \geq 4 \times 10^7$, $\epsilon$ is linearly decreased from 0.1 to 0.01; for $j > 4 \times 10^7$ we set $\epsilon = 0.01$. This strategy allows $\epsilon$ to stay above 0.01 for a fairly long time and facilitates exploration to mitigate the effects of local optima. We use this $\epsilon$ schedule in all of our experiments.

We set the learning rate to be $6.25 \times 10^{-5}$ following Hessel et al. (2018) unless otherwise specified. We use gradient clipping in the gradient descent iterations, using the maximal $\ell 2$ norm of 10 in Sec. 5.1 and 5.2, and 5 in Sec. 5.3. The $\epsilon_a$ hyperparameter for the Adam optimizer follows Hessel et al. (2018) and is set to be $1.5 \times 10^{-4}$ in Sec. 5.2, but in Sec. 5.1 it is set to be $1.5 \times 10^{-4}$ for DQN, and $5 \times 10^{-5}$ for C-DQN and $5 \times 10^{-6}$ for RG, becaue we observe that the sizes of the gradients are different for DQN, C-DQN and RG. $\epsilon_a$ is set to be $10^{-6}$ in Sec. 5.3 and E.3. The weight parameters in the neural networks are initialized following He et al. (2015), and the bias parameters are initialized to be zero.

### E.2 Prioritized sampling

In our experiments we have slightly modified the original prioritized sampling scheme proposed by Schaul et al. (2015). In the original proposal, in gradient descent optimization, a transition data $d_i = (s_t, a_t, r_t, s_{t+1}) \in \mathcal{S}$ is sampled with priority $p_i$, which is set to be

$$p_i = (|\delta_i| + \epsilon_p)^\alpha, \tag{33}$$

where $\epsilon_p$ is a small positive number, $\alpha$ is a sampling hyperparameter, and $|\delta_i|$ is the evaluated Bellman error when $d_i$ was sampled last time in gradient descent, which is

$$|\delta_{i(DQN)}| = \left| Q_\theta(s_t, a_t) - r_t - \gamma \max_{a'} Q_{\tilde{\theta}}(s_{t+1}, a') \right| \tag{34}$$

for DQN and

$$|\delta_{i(RG)}| = \left| Q_\theta(s_t, a_t) - r_t - \gamma \max_{a'} Q_\theta(s_{t+1}, a') \right|, \tag{35}$$

$$|\delta_{i(C\text{-}DQN)}| = \max \left\{ |\delta_{i(DQN)}|, |\delta_{i(RG)}| \right\}, \tag{36}$$

as we have chosen for RG and C-DQN, respectively. The probability for $d_i$ to be sampled is $P_i = \frac{p_i}{\sum_j p_j}$. To correct the bias that results from prioritized sampling, an importance sampling weight $w_i$ is multiplied to the loss computed on $d_i$, which is given by

$$w_i = \left( \frac{\sum_j p_j}{N} \cdot \frac{1}{p_i} \right)^\beta, \tag{37}$$

where $N$ is the total number of data and $\beta$ is an importance sampling hyperparameter. The bias caused by prioritized sampling is fully corrected when $\beta$ is set to be 1.

Schaul et al. (2015) propose using $\tilde{w}_i := \frac{w_i}{\max_j w_j}$ instead of $w_i$, so that the importance sampling weight only reduces the size of the gradient. However, we find that this strategy would make the learning highly dependent on the hyperparameter $\epsilon_p$ in Eq. (33), because given a data with vanishingly small $|\delta_i|$, its corresponding priority is $p_i \approx \epsilon_p^\alpha$, and therefore the term $\max_j w_j$ becomes $\max_j w_j \approx \left( \frac{\sum_j p_j}{N} \cdot \epsilon_p^{-\alpha} \right)^\beta \propto \epsilon_p^{-\alpha\beta}$. As a result, the gradient in learning is scaled by the term $\max_j w_j$ which is controlled by $\alpha$, $\beta$ and $\epsilon_p$, and $\max_j w_j$ changes throughout training and typically increases when the average of $|\delta_i|$ becomes large. For a given $\epsilon_a$ hyperparameter in the Adam optimizer, the overall decrease of the gradient caused by $\max_j w_j$ is equivalent to an increase of $\epsilon_a$, which effectively anneals the size of the update steps of the gradient descent. This makes $\epsilon_p$ an important learning hyperparameter, as also noted by Fujimoto et al. (2020), although this hyperparameter has been ignored in most of the relevant works including the original proposal. The results on *Space Invaders* for different values of $\epsilon_p$ are plotted in Fig. 14, which use the experimental settings in Sec. 5.1. It can be seen that the performance is strongly dependent on $\epsilon_p$. This issue may partially explain the difficulties one usually encounters when trying to reproduce published results.

**Lower bounded prioritization** To remove this subtlety, we use the original importance sampling weight $w_i$ instead of $\tilde{w}_i$. As $|\delta_i|$ is heavily task-dependent, to remove the dependence of $p_i$ on $\epsilon_p$ for all the tasks, we make use of the average $\bar{p} := \frac{\sum_j p_j}{N}$ to bound $p_i$ from below instead of simply using $\epsilon_p$ so as to prevents $p_i$ from vanishing. Specifically, we set $p_i$ to be

$$p_i = \max \left\{ (|\delta_i| + \epsilon_p)^\alpha, \frac{\bar{p}}{\tilde{c}_p} \right\}, \tag{38}$$

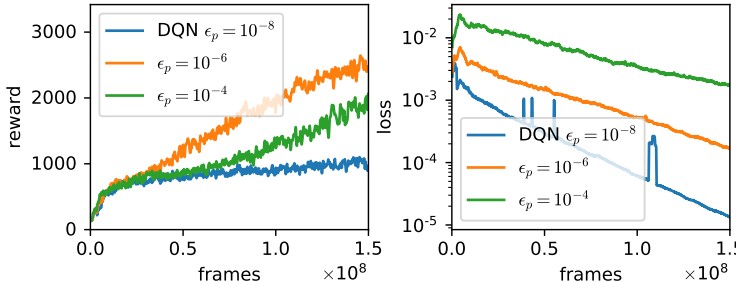

Figure 14: Training performance and loss for DQN on *Space Invaders*, with different hyperparameters $\epsilon_p$ and following the prioritization scheme in Schaul et al. (2015). The loss is calculated by multiplying $\tilde{w}_i$ and $\ell_{DQN}$ for each sampled data.

where we set $\epsilon_p$ to be a vanishingly small number $10^{-10}$, and $\tilde{c}_p > 1$ is a prioritization hyperparameter that controls the lower bound relative to the average. In our experiments we set $\tilde{c}_p = 10$. This scheme makes sure that regardless of the size of $|\delta_i|$, a data is always sampled with a probability that is at least $\frac{1}{\tilde{c}_p N}$, and $w_i$ is bounded by $\tilde{c}_p^\beta$ from above provided that the total priority $\sum_j p_j$ does not change too quickly. We adopt this prioritization scheme in all of our experiments except for the experiments in Fig. 14 above. Compared to Fig. 14, it can be seen that our DQN loss and C-DQN loss on *Space Invaders* in Fig. 4 do not change as much during training.

**Details of setting** When a new data is put into the replay memory, it is assigned a priority that is equal to the maximum of all priorities $p_i$ in the memory that have been calculated using Eq. (38), and at the beginning of learning when gradient descent has not started, we assign the collected data a priority equal to 100. We also additionally bound $w_i$ from above by $2\tilde{c}_p$, so that $w_i$ does not become too large even if the total priority $\sum_j p_j$ fluctuates. The hyperparameter $\beta$ is linearly increased from 0.4 to 1 during the $5 \times 10^7$ steps of the agent, following Schaul et al. (2015), and $\alpha$ is 0.9 in Sec. 5.3 and E.3 and is 0.6 otherwise. We did not try other values of $\alpha$.

In an attempt to improve efficiency, whenever we use $|\delta_i|$ to update the priority $p_i$ of a transition $(s_t, a_t, r_t, s_{t+1})$, we also use its half $\frac{|\delta_i|}{2}$ to compute $(\frac{|\delta_i|}{2} + \epsilon_p)^\alpha$, and use it as a lower bound to update the priority of the preceding transition $(s_{t-1}, a_{t-1}, r_{t-1}, s_t)$. This accelerates learning by facilitating the propagation of information. We adopt this strategy in all of our experiments except in Sec. 5.2 and in Fig. 14, in order to make sure that the results are not due to this additional strategy.

### E.3 EVALUATION OF THE DISCOUNT FACTOR AND NORMALIZATION OF THE LEARNED Q FUNCTION

**Evaluation of the discount factor** As discussed in Sec. B.5, some tasks require a large discount factor $\gamma$, while for many other tasks, a large $\gamma$ slows down the learning significantly and make the optimization difficult. Therefore, we wish to find a method to automatically determine a suitable $\gamma$ for a given task. As an attempt, we propose a heuristic algorithm to approximately estimate the frequency of the reward signal in an episode, based on which we can determine $\gamma$. The algorithm is described in the following.

Given an episode $\mathcal{E}_k$ in the form of an ordered sequence of rewards $\mathcal{E}_k = (r_i)_{i=0}^{T_k-1}$, for which $s_{T_k}$ is a terminal state,[8] we wish to have an estimate of the average number of steps needed to observe the next non-negligible reward when one starts from $i = 0$ and moves to $i = T_k$. Suppose that all rewards $\{r_i\}$ are either 0 or a constant $r^{(1)} \neq 0$; then, one may simply count the average number of steps before encountering the next reward $r^{(1)}$ when one moves from $i = 0$ to $T_k$. However, such a simple strategy does not correctly respect the different kinds of distributions of rewards in time, such as equispaced rewards and clustered rewards, and this strategy is symmetric with regard to the time reversal $0 \leftrightarrow T_k$, which is not satisfactory.[9] Therefore, we use a weighted average instead. To

---

[8] We consider $r_t$ to be the reward that is obtained when moving from state $s_t$ to $s_{t+1}$.

[9] This is because if one almost encounters no reward at the beginning but encounters many rewards at the end of an episode, the time horizon for planning should be large; however, if one encounters many rewards at the beginning but almost encounters no reward at the end, the time horizon is not necessarily large. Therefore,

this end we define the return

$$R_i(\mathcal{E}_k) := \sum_{t=i}^{T_k-1} r_t, \tag{39}$$

which is the sum of the rewards that are to be obtained starting from time step $i$, and we compute the average number of steps to see the next reward weighted by this return as

$$\hat{l}(\mathcal{E}_k) = \frac{\sum_{i=0}^{T_k-1} R_i(\mathcal{E}_k)\, l_i}{\sum_{i'=0}^{T_k-1} R_{i'}(\mathcal{E}_k)}, \qquad l_i := \min\{t \mid t \geq i, r_t = r^{(1)}\} - i + 1 \tag{40}$$

where $l_i$ is the number of time steps to encounter the next reward starting from the time step $i$. This strategy does not respect the time reversal symmetry as it involves $R_i$, and as the sum is taken over the time steps $\{0, 1, ...T_k - 1\}$, it can distinguish between clustered rewards and equispaced rewards, and it also properly takes the distance between consecutive clusters of rewards into account.

The next problem is how we should deal rewards that have various different magnitudes. As we can deal with the case of the rewards being either 0 or a constant, we may decompose a trajectory containing different magnitudes of rewards into a few sub-trajectories, each of which contains rewards that are either 0 or a constant, so that we can deal with each of these sub-trajectories using Eq. (40), and finally use a weighted average of $\hat{l}$ over those sub-trajectories as our result. With a set of episodes $\{\mathcal{E}_k\}$, we treat each of the episodes separately, and again, use a weighted average over the episodes as our estimate. The algorithm is given by Alg. 1. The weights in line 3 in Alg. 1 ensure that when computing the final result, episodes with a zero return can be ruled out, and that an episode with a large return does not contribute too much compared with the episodes with smaller returns. We have intentionally used the inverse in line 10 in Alg. 1 and used the root mean square in line 12 in order to put more weight on episodes that have frequent observations of rewards. Taking the absolute values of rewards in line 2 is customary, and in fact, one can also separate the negative and positive parts of rewards and treat them separately. Note that the output $f$ should be divided by a factor of 2 to correctly represent the frequency of rewards. We also notice that the above approach can actually be generalized to the case of continuous variables of reward and time, for which integration can be used instead of decomposition and summation.

To obtain the discount factor $\gamma$, we set the time horizon to be $\frac{\tilde{c}_\gamma}{f}$, with a time-horizon hyperparameter $\tilde{c}_\gamma$, and then we set $\gamma = 1 - \frac{f}{\tilde{c}_\gamma}$. To make $\gamma$ close to 0.9998 for the difficult games discussed in Sec. 5.3, we have set $\tilde{c}_\gamma = 15$. However, later we noticed that the agent actually learns more efficiently with a smaller $\gamma$, and $\tilde{c}_\gamma$ may be set to range from 5 to 15. In Sec. 5.3, $\tilde{c}_\gamma = 15$ is used, but in the following experiments we use $\tilde{c}_\gamma = 10$.

We make use of the complete episodes in the initial 50000 transitions collected at the beginning of training to evaluate $\gamma$, and here we also regard the loss of a life in the game as the end of an episode. We clip the obtained $\gamma$ so that it lies between 0.99 and 0.9998, and $\gamma$ is set to be 0.9998 if no reward is observed.

**Normalization of the Q function**    In addition to using the transformation function $\mathcal{T}(\cdot)$ to squash the Q function as described in Eq. (28) and (30), we normalize the Q function by the scale of the reward since the tasks in *Atari 2600* have vastly different magnitudes of rewards. For an episode $\mathcal{E}_k$, the Q function, or the value function, as the discounted return in the episode is given by

$$Q_{i;k} = \sum_{t=i}^{T_k-1} \gamma^{t-i} r_t, \tag{41}$$

and we compute its mean $\mu$ by taking the average of $Q_{i;k}$ over all states in given sample episodes. The standard deviation of $Q_{i;k}$, however, has a dependence on $\tilde{c}_\gamma$. If we simply normalize $Q_{i;k}$ by its standard deviation, the magnitude of the reward signal after normalization becomes dependent on the hyperparameter $\tilde{c}_\gamma$, which we wish to avoid. To obtain a normalization that is independent of $\tilde{c}_\gamma$, we assume that rewards are i.i.d. variables with mean $\mu_r$ and standard deviation $\sigma_r$. Focusing on

---

the discount factor should not be evaluated by a method that respects the time reversal. Note that the Fourier transform also respects the time reversal and thus does not suffice for our purpose.

---

**Algorithm 1** Estimation of the expected frequency of observing a next reward signal

**Input:** sample episodes $\{\mathcal{E}_k\}$
**Output:** an estimate of the inverse of the number of time steps $f$

1: **for** $\mathcal{E}_k \in \{\mathcal{E}_k\}$ **do**
2:     $\mathcal{E}_k = (r_i)_{i=0}^{T_k-1} \leftarrow (|r_i|)_{i=0}^{T_k-1}$                 ▷ Taking the absolute value of reward
3:     $w_k \leftarrow \sqrt{R_0(\mathcal{E}_k)}$                 ▷ For computing a weighted average over different episodes
4:     $j \leftarrow 0$
5:     **while** rewards $\mathcal{E}_k = (r_i)_{i=0}^{T_k-1}$ are not all zero **do**
6:         $j \leftarrow j + 1$
7:         $\mathcal{E}_k^{(j)}, \mathcal{E}_k \leftarrow \text{DECOMPOSESEQUENCE}(\mathcal{E}_k)$
8:     **end while**
9:     Compute $l_k \leftarrow \frac{\sum_j R_0(\mathcal{E}_k^{(j)}) \, \hat{l}(\mathcal{E}_k^{(j)})}{\sum_{j'} R_0(\mathcal{E}_k^{(j')})}$ ▷ Weighting the results by the contribution of the rewards
10:     $f_k \leftarrow \frac{1}{l_k}$
11: **end for**
12: Compute $f \leftarrow \sqrt{\frac{\sum_k w_k f_k^2}{\sum_{k'} w_{k'}}}$     ▷ We use RMS to have more emphasis on episodes with larger $f_k$
13: **return** $f$
14:
15: **procedure** DECOMPOSESEQUENCE($\mathcal{E}$)
16:     $r' \leftarrow \min\{r_i\}_{i=0}^{T-1} = \min \mathcal{E}$
17:     $\mathcal{E}' \leftarrow (r_i'')_{i=0}^{T-1}, \quad r_i'' := \begin{cases} 0, & \text{if } r_i = 0 \\ r', & \text{otherwise} \end{cases}$
18:     $\mathcal{E} \leftarrow (r_i - r_i'')_{i=0}^{T-1}$
19:     **return** $\mathcal{E}', \mathcal{E}$
20: **end procedure**

---

the Q function at the initial states, i.e. $Q_{0;k}$, we obtain the relation

$$\mathbb{E}[Q_{0;k}] = \frac{1 - \gamma^{T_k}}{1 - \gamma} \mu_r, \tag{42}$$

and therefore we estimate $\mu_r$ by $\mu_r = \frac{1}{N_\mathcal{E}} \sum_k Q_{0;k} \frac{1-\gamma}{1-\gamma^{T_k}}$, with the number of sample episodes $N_\mathcal{E}$. Also, we have

$$Var\left(Q_{0;k} - \frac{1 - \gamma^{T_k}}{1 - \gamma} \mu_r\right) = \frac{1 - \gamma^{2T_k}}{1 - \gamma^2} \sigma_r^2, \tag{43}$$

and therefore $\sigma_r$ can be estimated by the variance of $\left\{\left(Q_{0;k} - \frac{1-\gamma^{T_k}}{1-\gamma}\mu_r\right) \cdot \sqrt{\frac{1-\gamma^2}{1-\gamma^{2T_k}}}\right\}_k$.

After obtaining the standard deviation of rewards $\sigma_r$, we need to compute a scale $\sigma$ to normalize the learned Q function. To avoid $\tilde{c}_\gamma$ dependence of $\sigma$, we use $\sigma_r$ to roughly predict the standard deviation of $Q_{0;k}$ if $\gamma_0 \equiv 1 - \frac{f}{2}$ is used as the discount factor, and we use the obtained standard deviation as the normalization factor $\sigma$. For simplicity we ignore the variance of $T_k$ and obtain

$$\sigma = \sigma_r \cdot \frac{1}{N_\mathcal{E}} \sum_k \sqrt{\frac{1 - \gamma_0^{2T_k}}{1 - \gamma_0^2}}. \tag{44}$$

Finally, the function we let the agent learn is $\hat{Q} \equiv \frac{Q-\mu}{\sigma}$.

The normalized function $\hat{Q}$ also obeys the Bellman equation as well, but with a slightly modified reward. It can be easily shown that

$$\hat{Q}^*(s_t, a_t) = \frac{r_t - (1 - \gamma)\mu}{\sigma} + \gamma \max_{a'} \hat{Q}^*(s_{t+1}, a') \quad \text{if } s_{t+1} \text{ is non-terminal}, \tag{45}$$

and

$$\hat{Q}^*(s_t, a_t) = \frac{r_t - \mu}{\sigma} \quad \text{if } s_{t+1} \text{ is terminal}. \tag{46}$$

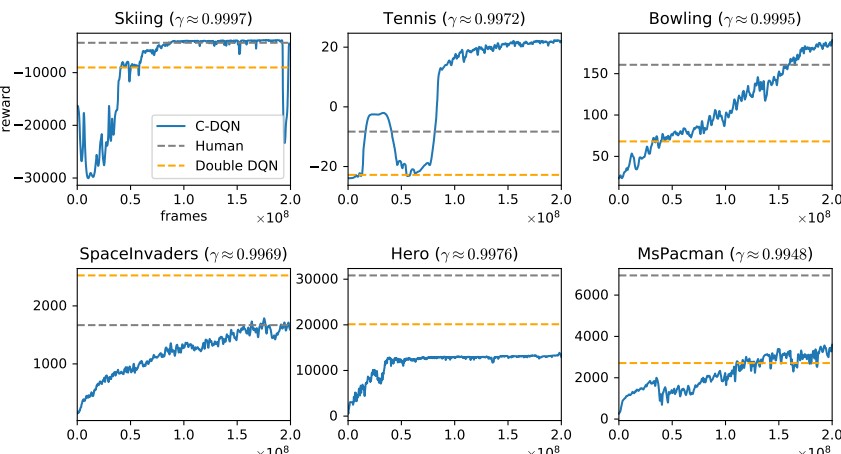

Figure 15: Training performance for C-DQN on several games in *Atari 2600* compared with the human performance (Badia et al., 2020) and the double DQN (Hessel et al., 2018), using the same experimental setting as in Sec. 5.3, except for using $\tilde{c}_\gamma = 10$.

Therefore, the effect of normalization amounts to modifying the reward $r_t$ to $\hat{r}_t := \frac{r_t-(1-\gamma)\mu}{\sigma}$, and then assigning an additional terminal reward $-\frac{\gamma\mu}{\sigma}$. This is easy to implement and we use this normalization in our experiments in Sec.5.3. Similarly to the evaluation of $\gamma$, we use the episodes in the initial 50000 transitions to compute $\mu$ and $\sigma$; however, here we do not regard the loss of a life as the end of an episode, so that the lengths of the episodes do not become too short. We also do not take episodes that have a zero return into account.

**Results on other Atari games** To demonstrate the generality of the above strategy, we report our results on several games in *Atari 2600*, using a learning rate of $4 \times 10^{-5}$ and $\tilde{c}_\gamma = 10$. The results are shown in Fig. 15. We find that for some games, especially *Hero*, the learning sometimes gets trapped in a local optimum and learning may stagnate, which deserves further investigation. We did not do a fine grid search on the learning rate and we simply selected from $6 \times 10^{-5}$, $4 \times 10^{-5}$ and $2 \times 10^{-5}$, and we chose $4 \times 10^{-5}$, because it produces reasonable results for most of the tasks. We notice that it is difficult to find a learning rate that works well for all the tasks, as the tasks have drastically different levels of stochasticity and are associated with different time horizons.

We also notice that there are several cases where our strategy of the normalization and the evaluation of $\gamma$ does not give satisfactory results. This is mainly because we have assumed that the time scales of obtaining rewards are similar for the random policy and for a learned policy. A typical counterexample is the game *Breakout*, where the random policy almost obtains no reward in an episode but a learned policy frequently obtains rewards. Therefore, our strategy above is still not general enough to deal with all kinds of scenarios, and it cannot replace the bandit algorithm in Badia et al. (2020) which is used to select $\gamma$, and therefore a better strategy is still desired.

## F EXPERIMENTAL DETAILS ON CLIFF WALKING

In the cliff walking experiments, we store all state-action pairs into a table, excluding the states of goal positions and cliff positions, and excluding the actions that go into the walls. We plot the data on a log scale of iteration steps by explicitly evaluating the loss over all state-action pairs in Sec. 3.1, and evaluating the reward of the greedy policy in Sec. 3.2. Because we do evaluations at equal intervals on a log scale of the x-axis, fewer evaluations are made when the number of iteration steps is larger, and as a consequence, the scatter plots in the right of Fig. 1 do not have equally many data points along the curves. The learning rate $\alpha$ is always 0.5, and $\epsilon$ in the $\epsilon$-greedy policy is always fixed. Specifically for the one-way cliff walking task in Fig. 3, when two actions $a_1$ and $a_2$ have the same Q function value, i.e. $Q(s_t, a_1) = Q(s_t, a_2)$, the greedy policy randomly chooses $a_1$ or $a_2$ at state $s_t$, and when $\max_{a'} Q(s_{t+1}, a') = Q(s_{t+1}, a_1) = Q(s_{t+1}, a_2)$, we modify the learning rule

of RG for transition $(s_t, a_t, r_t, s_{t+1})$ to be

$$
\Delta Q(s_t, a_t) = \alpha \left( r_t + \gamma \max_{a'} Q(s_{t+1}, a') - Q(s_t, a_t) \right),
$$

$$
\Delta Q(s_{t+1}, a_1) = \Delta Q(s_{t+1}, a_2) = -\frac{\gamma}{2} \Delta Q(s_t, a_t), \tag{47}
$$

so that the greedy policy at $s_{t+1}$ is not changed after learning from the transition.

