# OpenReview forum: "Convergent and Efficient Deep Q Learning Algorithm"
_ICLR.cc/2022/Conference — ICLR 2022 Poster_

### Official Review · Reviewer_x6bM · 2021-10-21

**Correctness:** 4
**Technical Novelty And Significance:** 2
**Empirical Novelty And Significance:** 2
**Recommendation:** 6
**Confidence:** 3

**Main Review:**

Overall, I feel this paper is well-organized and well-written. I can understand the message that the authors want to deliver. The experiments are well-designed and clean. The main limitation from my point of view is that there seems to be not too much new stuff.

The only change w.r.t to DQN is Line 12 which modifies the loss to be the maximum of DQN loss and MSBE loss. All the other components remain the same and all the drawbacks are inherited (sample-inefficiency and exploration). A lot of efforts are made on some corner cases that DQN and MSBE fail. Since there is no theory for C-DQN (I mean convergence is not the ultimate goal since it may converge to a bad local solution. For online RL, you should study regret. For policy optimization, you should study sample-complexity), I am wondering if there exist other corner cases that C-DQN could also suffer.

When the paper mentioned efficiency or inefficiency, I think it is worth pointing out it is sample-efficiency or computational efficiency. For the problem pointed out in Section 3, how it relates to the problem of exploration? Because from the description, the data quality seems to play an important role. I admit the exploration for deep RL is hard, but at least for tabular cases, there are sample and computationally efficient algorithms, say https://arxiv.org/abs/1807.03765 Is Q-learning Provably Efficient?. The community may need some principle ways to understand the inefficiency. Current Section 3.2 seems to be not very in principle.

The limitation of MSBE has been studied for a long time in RL community. There are multiple important improvements on this but the authors didn't discuss in detail (although the authors cited them in one sentence). That means this paper is not properly positioned in literature. The widely known issue is the double-sampling issue (empirical MSBE is biased) and several works have been proposed to remedy this, "A Kernel Loss for Solving the Bellman
Equation". But in this paper, the authors still discussed and compared the original version of RG algorithm. Since the discussion in Section 3.1 is quite specific, it's hard for me to know if a similar limitation holds for recently proposed algorithms.

"Although convergent gradient-based methods have also been proposed (Sutton et al., 2009; Bhatnagar et al., 2009; Feng et al., 2019; Ghiassian et al., 2020), they cannot be used with neural networks easily, and they often have worse performance than TD methods and DQN." If you want to make such arguments, please provide detailed references and explain the reason.

"In this work, we only consider the deterministic case and drop the notation Est+1 [·] where appropriate." I am wondering if the whole paper is limited to deterministic transition. If so, I think this should be highlighted in the abstract. In theory, there is a huge difference between deterministic case and stochastic transition.

Minor: 1. "If the task is difficult" is vague. What do you mean by a difficult task? 2. In Figure 4, what the loss is for?


**Summary Of The Paper:**

This paper studies several issues of DQN (basically FQI, a version of fixed-point iteration) and empirical Bellman error minimization (has a clear loss function). Several issues of MSBE are pointed out. The authors propose a new approach by replacing the DQN loss by the maximum of DQN loss and MSBE loss. Several experiments on Atari games support the argument.

**Summary Of The Review:**

The noverlty and significance of the proposed algorithm is limited.

---

> ### Author Response · Authors · 2021-11-19
> **Reply to Reviewer x6bM: Part 1**
>
> We gratefully thank you for careful reading and your pertinent questions and criticisms. We apologize for the lack of clarity regarding the relation between our work and previous works. Our C-DQN strategy is proposed as a minimal modification to the conventional DQN so that the algorithm is guaranteed to converge and does not suffer from instability and divergence issues in practical applications. As commented by reviewer bKTr, the simplicity is actually an advantage of our method, because the simplicity allows our method to be "incorporated into a wide variety of (probably even most) Q-function based RL procedures without much effort" and to benefit those strategies by improving their stability. In the following, we present our reply to the questions and issues you raised.
>
> Regarding the underlying theory of C-DQN, we have to admit that we currently do not have rigorous theoretical results on the efficiency or complexity of the general algorithm. Nevertheless, as C-DQN is based on DQN and RG, many of the results on DQN and RG are supposed to apply to C-DQN as well. Especially, if one considers online learning for deterministic tabular problems, C-DQN reduces to Q-table learning. This is because the idea of C-DQN is to prevent the Bellman residual of a transition $(s_t,a_t,r_t,s_{t+1})$ from increasing after performing the Q-learning update step, and therefore, in the tabular case, C-DQN is trivial and is reduced to Q-table learning since the Bellman residual is not increased. Also, as the loss of C-DQN is larger than or equal to MSBE, optimality bounds for MSBE apply to C-DQN as well. Therefore, the underlying principle of C-DQN is closely related to that of Q-learning and RG, and C-DQN may be regarded as a technical modification to the DQN learning rule to prevent divergence caused by function approximation. Specifically, as MSBE is equal to the DQN loss when the target network is updated, C-DQN effectively follows the DQN loss while making sure that the DQN loss is not increased when the target network is updated next time. Therefore, the underlying theory of DQN and that of C-DQN may be considered as similar. We admit that there might be some corner cases in which C-DQN fails; however, so far we are not aware of any of them, and as we have verified the validity of C-DQN on the diverse and general \textit{Atari 2600} benchmark which includes different types of games played by human, we believe that C-DQN is robust and general enough to serve as a convergent alternative to the conventional DQN method.
>
> Regarding the issues of inefficiency discussed in Section 3, in principle, the ill-conditionedness issue in Section 3.1 which leads to computational inefficiency applies to all recent improvements on RG that we have mentioned. This is because almost all recent improvements have focused on the double-sampling issue and have essentially replaced the double-sampled value by an estimated or learned one, so as to reproduce the unbiased gradient obtained by double sampling in expectation. Therefore, if the system is deterministic, ideally, the estimated value should be identical to the gradient that is directly calculated in RG, and all those methods mentioned in our paper are supposed to reduce to the RG method for deterministic tabular problems. Specifically, the saddle-point formulation essentially learns the expected reward starting from a state $s_t$, and then calculates the gradient using a single sample $s_{t+1}$; the kernel method uses the Bellman error of states that are similar to $s_t$ to replace the double sampling and estimate the unbiased gradient, where the similarity is measured by the kernel, as can be seen in Eq.~(4) in the work ``A Kernel Loss for Solving the Bellman Equation''. Therefore, all these methods are supposed to be the same and not better than RG if the underlying task is deterministic. On the other hand, our work focuses on completely different problems associated with RG, which have not been investigated in previous works. We find that RG cannot learn effectively even when there is no double sampling issue due to the problems in its learning dynamics as discussed in Section 3 in our work.

---

> > ### Author Response · Authors · 2021-11-19
> > **Reply to Reviewer x6bM: Part 2**
> >
> > For the issue of inefficiency in Section 3.2, the reason why exploration is important is because in online learning, without additional exploration strategies, RG cannot obtain the data that are necessary for it to learn the optimal solution, because RG does not consistently learn to move to the states that produce more reward, as we have demonstrated using toy tasks. The inefficiency discussed in the section assumes the use of the $\epsilon$-greedy policy, because it is the most common exploration strategy used in Q-function based deep RL, and other exploration strategies in deep RL are often considerably more complicated (e.g.~involving multiple learning procedures) and are designed for Q-learning instead of RG. Therefore, although there might exist certain exploration strategies for RG to search for better data to learn, as we aim to show the inefficiency of RG in the standard scenario of deep RL, we have considered the $\epsilon$-greedy policy only and discussed within this context. We agree that the discussion in Section 3.2 is not very formal or, ``in principle'', and unfortunately, we are not aware of any other works that have discussed a similar problem, and we cannot easily fit our result into conventional formalisms such as regret or complexity. For example, for the issue discussed in the first part of Section 3.2, with $\gamma=1$, RG cannot learn regardless of how much computation or how many samples it uses, although all data are uniformly sampled to train the agent. This occurs because the loss of RG is trapped on a plateau where the gradient is constantly zero, and therefore the learning cannot proceed. As these issues are dependent on the structure of the state space, the initialization of the Q function and the magnitude of the rewards, it seems difficult to give a precise and quantitative description of the issues, and currently we can only present an intuitive picture to discuss the failure modes observed on RG. We believe that the issues discussed in our work can stimulate further research on the problems in learning dynamics and we hope the issues can be formally settled in future works.
> >
> > Our reply to the other questions are presented in the following.
> >
> > >"Although convergent gradient-based methods have also been proposed (Sutton et al., 2009; Bhatnagar et al., 2009; Feng et al., 2019; Ghiassian et al., 2020), they cannot be used with neural networks easily, and they often have worse performance than TD methods and DQN." If you want to make such arguments, please provide detailed references and explain the reason.
> >
> > The reason is that these methods either require the linearity of the learned function, or require computationally heavy operations such as computing the Hessian matrix, which cannot be used efficiently with large neural networks. We have revised our text to clarify this problem. To our knowledge, our proposed C-DQN is the first convergent Q-function based method that is scalable and can be efficiently used with large neural networks to solve complicated tasks such as the \textit{Atari 2600} benchmark.
> >
> > >"In this work, we only consider the deterministic case and drop the notation Est+1 [·] where appropriate." I am wondering if the whole paper is limited to deterministic transition. If so, I think this should be highlighted in the abstract. In theory, there is a huge difference between deterministic case and stochastic transition.
> >
> > Although the discussion presented in the main text of the paper focuses on deterministic transition, the proposed C-DQN is found to work satisfactorily and robustly for both deterministic and stochastic problems, and its convergence property is not affected. We find that the C-DQN algorithm converges between the solutions found by DQN and RG in general, as discussed in Section A in our revised manuscript.
> >
> > >Minor: 1. "If the task is difficult" is vague. What do you mean by a difficult task? 2. In Figure 4, what the loss is for?
> >
> > We have revised the text and deleted the sentence "the task is difficult" to avoid unnecessary confusion. Here, by a "difficult task" we have meant a task that has complicated structures, e.g. given two states with similar representations the Q function values can be vastly different, and the space of the state is large and complex, such as the game Go. In Figure 4, the loss functions are the DQN loss, MSBE and the C-DQN loss. The loss functions are given by Eqs. (10-12) and (5).
> >
> > We believe that our reply has successfully answered your questions and criticisms, and we do hope our work can meet with your approval.

---

> > > ### Comment · Reviewer_x6bM · 2021-11-25
> > > **Increase the score**
> > >
> > > Thanks for the detailed response. Based on that, I increase my score. After reading the paper again, I feel the ill-conditionness section is good and also please make it more clear that you are targeting for the different issues of MSBE than double-sampling. For Section 3.2, it looks like the authors admit that exploration is the problem. But it's pretty well-known that epsilon-greedy is inefficient combining with almost any planning method. You can easily come up with a bandit example which is a strictly sub-class of MDP. Why you spend so much effort explaining the exploration problem for the specific MSBE algorithm? I feel even though you modify the loss, you will still suffer the original exploration issue (data quality) caused by epsilon-greedy.

---

> > > > ### Author Response · Authors · 2021-11-27
> > > > **Reply to Reviewer x6bM (2)**
> > > >
> > > > We thank you for your appreciation of our work and your question. In fact, deep RL, especially deep Q-learning, often works satisfactorily with $\epsilon$-greedy as the only exploration strategy. This is because in many commonly encountered cases, the agent following the optimal policy always obtains short-term reward frequently along its trajectory, and therefore, the policy that maximizes short-term reward is close to the optimal policy, which is a commonly used heuristic. Then, in order to find the optimal policy, the agent only needs to explore for short-term reward starting from each of its frequently visited states, reducing the required depth of search and making it unnecessary to search exponentially in the size of the system. This is evident in our cliff-walking example in Section 3.2, in which case the agent only needs to search for a single step. However, even in this case, RG cannot learn efficiently as it does not follow the path that leads to more reward, and one has to use additional exploration to force the agent to repeatedly visit highly rewarding states. This is the case for most of the tasks in the Atari 2600 benchmark, for which short-term search works sufficient well, although the efficiency of the exploration is usually not high.
> > > >
> > > > We do hope that we have successfully addressed your question.

---

### Official Review · Reviewer_bKTr · 2021-10-30

**Correctness:** 4
**Technical Novelty And Significance:** 3
**Empirical Novelty And Significance:** 3
**Recommendation:** 10
**Confidence:** 4

**Main Review:**

Strengths\
The topic is very relevant, with potentially very strong impact on the field of Q-function based RL.

The paper is excellently written.

The method presented is extremely simple and can be incorporated into a wide variety (probably even most) Q-function based RL procedures without much effort.

Weaknesses\
In my opinion, the choice of benchmarks is not well suited for examining limitations of the method. I suspect that in benchmarks that are strongly stochastic in the sense that from a state-action pair, successor states with strongly different values of the value function can be reached, the Bellman residual loss function dominates and therefore no advantage is gained over optimization with respect to the Bellman residual loss function and thus the solutions do not solve the RL problem.
It would, in my opinion, add considerable value to the already good paper if the behavior of the method were additionally examined on one or more benchmarks that have the property that from a state-action pair, successor states with strongly different values of the value function can be reached.
I realize that this cannot be easily incorporated into the limited number of pages, and the current excellent clarity of presentation should not suffer. Possibly this investigation could go into the appendix. Or this investigation could be considered as Future-Work.

Further comments\
I would appreciate it if in addition to current works that represent the field well, the pioneering works would also be referenced in each case. So instead of (Mnih et al., 2015; Hessel et al., 2018) rather (Riedmiller 2005, Mnih et al., 2015; Hessel et al., 2018).

Riedmiller 2005, Neural Fitted Q Iteration - First Experiences with a Data Efficient Neural Reinforcement Learning Method

And instead of (Munos & Szepesvári, 2008), rather (Ernst et al. 2005, Munos & Szepesvári, 2008).

Ernst et al. 2005, Tree-Based Batch Mode Reinforcement Learning

A simple benchmark where the property that successor states with widely varying values of the value function can be reached from a single state-action pair is pronounced is the wet chicken benchmark (e.g., Hans & Udluft 2011). There is a one-dimensional variant and a two-dimensional variant that is somewhat more difficult. Perhaps an investigation with this, or a similar benchmark, would eliminate the concerns that in this case the Bellman residual is always dominant and CDQN performs worse than DQN ... or confirm it.

Hans & Udluft 2011, Ensemble Usage for More Reliable Policy Identification in Reinforcement Learning

I am not sure what is meant by the term "overgeneralization". If this means something other than the following definition: \
*The American Psychological Association defines overgeneralization as, “a cognitive distortion in which an individual views a single event as an invariable rule, so that, for example, failure at accomplishing one task will predict an endless pattern of defeat in all tasks.”*
then this should be explained and/or supported by a reference.

In "we construct a loss does not" probably a "that" is missing.

In Table 1 in the Appendix is written "DQN (ours)".

Please check the bibliography for accidental lower case letters, like „bellman“, „q-“, „td“


**Summary Of The Paper:**

The paper addresses the problem of unstable learning that arises in NFQ type RL methods.
It is assumed that the problem can be overcome by a provably convergent NFQ-like method.
Such a provably convergent method is presented and tested on a selection of benchmarks and compared to DQN.


**Summary Of The Review:**

The paper is excellently written. The method presented could be groundbreaking. However, it is not explored whether the proposed method is generally applicable (see "Main Review" for details).

---

> ### Author Response · Authors · 2021-11-16
> **Reply to Reviewer bKTr**
>
> We gratefully acknowledge your careful reading and appreciation of our work and your valuable advice. As per your suggestion, we have reproduced the 1D wet-chicken experiment in Hans \& Udluft (2011) and compared the performance of C-DQN, DQN (NFQ) and RG. The detailed results are presented in the additional A.1 section in the appendix in our revised manuscript. The results show that while the performance of RG is significantly worse than the performance of DQN (NFQ), i.e., 11 versus 14.5, the performance of C-DQN is only slightly worse than DQN, i.e., 14 versus 14.5. Since the random policy already achieves the performance of 6.7, C-DQN performs significantly better than RG and behaves more similarly to DQN than RG. Therefore, according to this experimental result, C-DQN is robust and produces satisfactory results even when the task is highly stochastic, although it may not produce the optimal solution. In fact, we have expected this experimental result, because when the target network is updated and when the algorithm converges, both the Bellman residual loss and the DQN loss are equal, and therefore, neither of the losses should dominate over the other, and the learning should not be dominated by the Bellman residual loss simply because of the stochasticity in the task. Specifically, in stochastic cases, we have
> $$L_{\textit{MSBE}}(\theta)=\left (Q_\theta(s_t,a_t)-r_t-\gamma \mathbb{E}\_{s_{t+1}}\left[\max_{a'}Q_\theta(s_{t+1},a')\right]\right )^2 + \gamma^2\text{Var}\_{s_{t+1}}(\max_{a'}Q_\theta(s_{t+1},a'))$$
> and
> $$L_{\textit{DQN}}(\theta;\tilde{\theta})=\left (Q_\theta(s_t,a_t)-r_t-\gamma \mathbb{E}\_{s_{t+1}}\left[\max_{a'}Q_{\tilde{\theta}}(s_{t+1},a')\right]\right )^2 + \gamma^2\text{Var}\_{s_{t+1}}(\max_{a'}Q_{\tilde{\theta}}(s_{t+1},a')), $$
> and therefore the magnitudes of $L_{\textit{MSBE}}$ and $L_{\textit{DQN}}$ are similar, and the problematic part in $L_{\textit{MSBE}}$ is the term $\text{Var}\_{s_{t+1}}(\max_{a'}Q_\theta(s_{t+1},a'))$, which does not direct the gradient $\nabla_\theta L_{\textit{MSBE}}$ towards the solution of the Bellman equation. In contrast, the variance term in $L_{\textit{DQN}}$ is independent of $\theta$ and does not contribute to the gradient $\nabla_\theta L_{\textit{DQN}}$. Therefore, stochasticity does not make the magnitude of one loss dominate over the other, but makes the solutions found by RG and DQN differ, in which case C-DQN converges somewhere between the two solutions. In the wet-chicken experiment, we have additionally confirmed that the Q function learned by C-DQN indeed lies in between the Q function learned by DQN and that learned by RG, which supports our argument, as shown in section A.1.
>
> We would like to thank you for informing us of the pioneering works in this field, and we have checked and included the works you have mentioned in our references. We have corrected the typos and the mistakes in our formats you mentioned, and we have revised the manuscript and corrected several other minor mistakes and improved the readability. We noticed that the word "overgeneralization" is actually not widely recognized in RL and causes unnecessary confusion, and we have revised the text to avoid using the word "overgeneralization". According to our understanding, as "generalization" refers to the fact that learning one data point affects the prediction of the model for other data points, "overgeneralization" refers to the problem that the "generalization" can sometimes lead to undesirable consequences, such as interference among the learning of different data points.
>
> We do hope the new experimental results and our reply above successfully answer your questions and resolve the concerns, and we thank you again for your appreciation of our work and valuable advice.
>
>
>
> Hans \& Udluft 2011, Ensemble Usage for More Reliable Policy Identification in Reinforcement Learning

---

> > ### Comment · Reviewer_bKTr · 2021-11-20
> > **Excellent feedback**
> >
> > I am really impressed with the amount of effort and care the authors have put into responding to my concerns and suggestions. That is truly excellent. The good results on the wet-chicken benchmark very much strengthen my confidence that we are dealing with a significant improvement here. The paper should definitely be published in my opinion. The impact could be immense.

---

> > > ### Comment · Reviewer_bKTr · 2021-11-20
> > > **A revolution?**
> > >
> > > I feel like I should plausibilize my extremely positive score. For one thing, the between scores do not exist. So I can only choose 8 or 10 and not 9.
> > > And 8 seems too low to me, because the comparison on the wet-chicken benchmark, where RG must (and does) fail completely, shows quite clearly that the proposed method really works, thus giving new hope to having a convergent algorithm that doesn't aim for the useless RG solution, but instead gives new impetus to the correct TD solution. Sure, I may be missing something here, but if this is true, and works like that, it could spark a revolution. Really.

---

> > > ### Author Response · Authors · 2021-11-22
> > > **Reply to Reviewer bKTr (2)**
> > >
> > > We really thank you for your appreciation and your kind reply. Again, we would like to thank you for informing us of the wet-chicken benchmark, which has helped us to investigate and compare DQN, C-DQN and RG in a stochastic and well-understood setting and has greatly improved our work. We believe that the proposed C-DQN method can help to solve many problems which could not be solved using conventional methods due to instability issues, and that it can open up new possibilities for the field of RL in general.

---

### Official Review · Reviewer_SCA5 · 2021-11-01

**Correctness:** 4
**Technical Novelty And Significance:** 3
**Empirical Novelty And Significance:** 2
**Recommendation:** 6
**Confidence:** 4

**Main Review:**

The primary way in which the authors approach the problem of convergence of DQN and related algorithms is replace the use of a DQN-style target network with a minimization of the mean squared Bellman error (MSBE), the foundation of the residual gradient-type algorithms introduced in 1995. This type of algorithms however did not traditionally perform better than other approaches, especially in the context of deep RL.

The authors are proposing an approach where the loss of their new algorithm C-DQN is the expectation of the maximum between the MSBE loss and a DQN loss which is derived from interpreting DQN as a form of fitted iteration. This definition enforces that both the DQN and the MSBE loss are decreasing during learning.

While this guarantee is interesting, the experimental results do not show a clear motivation for the use of this approach. The experiments in which the authors succeed to show an improvement are hand engineered to show off the benefits of the approach - for instance the authors compare against DQN on space invaders where half of the data is randomly discarded, or when the replay buffer is using a random replacement strategy.

**Summary Of The Paper:**

The paper argues that the DQN and its algorithm and its variants do not guarantee convergence and that they can diverge in realistic settings.

The authors develop a new technique that guarantees convergence of DQN. The authors show that in the case of very large discount factors (such as 0.9998) this algorithm outperforms baseline algorithms on certain Atari games.

**Summary Of The Review:**

The paper introduces an interesting perspective on guaranteed convergence for DQN. However, this is not the first algorithm that is doing so, and the experimental results show improvement only in specific selected, somewhat unusual circumstances.

---

> ### Author Response · Authors · 2021-11-22
> **Reply to Reviewer SCA5**
>
> We would like to thank you for your careful reading and pertinent criticisms, and we apologize for not presenting a clear motivation of our method. We agree that many of our experiments represent unusual circumstances; however, we believe that it is an important goal for general reinforcement learning to deal with all kinds of circumstances including unusual ones, and our work indeed contributes to the field of reinforcement learning significantly. In addition, to our knowledge, our proposed C-DQN algorithm is actually the first convergent method that is sufficiently scalable and efficient to work with large neural networks and obtains successful results on the standard Atari 2600 benchmark. We have revised the text in our manuscript to emphasize this fact for clarification.
>
> Moreover, the convergence property is indeed beneficial for DQN in practical scenarios, as DQN often requires fine-tuning or technical modifications to work stably. For example, as DQN shows instability and divergence in loss when the discount factor $\gamma$ is too large or when the update period of the target network is too short, one needs to fine-tune the hyperparameters so that DQN can learn stably and achieve satisfactory performance. However, a small $\gamma$ implies that the time horizon for planning cannot be long, and a long update period of the target network implies that the speed of learning cannot be fast, and therefore, one has to sacrifice efficiency and long-term planning of DQN in order to obtain stable performance, and in some cases, it can be extremely difficult to obtain satisfactory performance. One typical example is our experiment on the Atari game Skiing, which is elaborated in Section B.4 in the appendix. The game Skiing requires a discount factor that is larger than $0.999$ to learn, and the previous state-of-the-art result on this game was obtained by Agent 57 (Badia et al., 2020) by combining many sophisticated techniques and learning with several different $\gamma$ values simultaneously. Nevertheless, with C-DQN, we can surpass the performance of Agent 57 using less than 0.1\% of its computational budget by simply training with a larger discount factor. This shows the effectiveness of C-DQN for tasks that require long-term planning and large discount factors. In fact, the original DQN algorithm already shows divergent behaviour on several games in Atari 2600 with the standard hyperparameters, such as the games Wizard of Wor and Asterix, as shown in Van Hasselt et al. (2016). Although double Q-learning was proposed to suppress the overestimation of the Q values and has been shown to remove the divergence for these tasks, double Q-learning does not solve the issue of divergence in general, and therefore its effectiveness is only on a case-by-case basis. Pohlen et al. (2018) also introduces an additional temporal consistency loss to reduce the difference between the current network $Q_\theta$ and the target network $Q_{\tilde{\theta}}$ in order to stabilize the learning to use a larger discount factor. In practice, when one encounters divergence or instability issues, one often needs to do a significant amount of hyperparameter tuning and try different approaches to see whether the issue is solved or not, as there is no principled way to remove the divergence in existing literature and one has to deal with the problems on a case-by-case basis. In contrast, we propose C-DQN with guaranteed convergence to solve this issue. Therefore, as commented by reviewer bKTr, we believe that "the topic is very relevant, with potentially very strong impact on the field of Q-function based RL", and our work indeed contributes to the field of RL significantly.
>
> We believe that we have clarified the motivation of our work, and we do hope that our work can meet with your approval.
>
>
> References:
>
> Adrià Puigdomènech Badia, Bilal Piot, Steven Kapturowski, Pablo Sprechmann, Alex Vitvitskyi,
> Zhaohan Daniel Guo, and Charles Blundell. Agent57: Outperforming the atari human benchmark.
> In International Conference on Machine Learning, pp. 507–517. PMLR, 2020.
>
> Hado Van Hasselt, Arthur Guez, and David Silver. Deep reinforcement learning with double Q-learning. In Proceedings of the AAAI Conference on Artificial Intelligence, volume 30, 2016.
>
> Tobias Pohlen, Bilal Piot, Todd Hester, Mohammad Gheshlaghi Azar, Dan Horgan, David Budden, Gabriel Barth-Maron, Hado Van Hasselt, John Quan, Mel Vecerík, et al. Observe and look further: Achieving consistent performance on atari. arXiv preprint arXiv:1805.11593, 2018.

---

> > ### Comment · Reviewer_SCA5 · 2021-11-29
> > **Raised my review score to "marginally above"**
> >
> > Thank you for the detailed answer. I have raised the review score to "marginally above", considering that the paper might be interesting to some readers. Nevertheless, I still maintain that the performance had only improved on a very narrow set of examples selected to match the properties of the algorithm, so I cannot raise my ranking further.

---

### Decision · Program_Chairs · 2022-01-20

**Decision:**

Accept (Poster)

**Comment:**

It is important to have good stable and trustworthy algorithms.  Though I am unconvinved that the C-DQN algorithm proposed here is the final word (and I suppose this is not controversial, and the authors might agree), the ideas presented here are sufficiently interesting to be disseminated and discussed more widely.  All reviewers recommended accepting the paper, and I'll follow their lead.

That said, the paper can still be improved, and the authors are encouarged to carefully consider the feedback provided by the reviewers.  In particular, it is good to be clear about which parts are principled, and which parts are somewhat heuristic or arbitrary, and could therefore presumably be improved in future work.  In fact, doing so clearly could make the paper _more_ rather than less impactful.

In any case, it seems good to include this paper at the conference, to highlight the questions and partial answers given here, and to inspire more discussion.